# Localised surface plasmon resonance inducing cooperative Jahn–Teller effect for crystal phase-change in a nanocrystal

Masanori Sakamoto [1] ✉, Masaki Hada [2] ✉, Wataru Ota [3,4], Fumihiko Uesugi[5] & Tohru Sato[3,4]

The Jahn–Teller effect, a phase transition phenomenon involving the spontaneous breakdown of symmetry in molecules and crystals, causes important physical and chemical changes that affect various fields of science. In this study, we discovered that localised surface plasmon resonance (LSPR) induced the cooperative Jahn–Teller effect in covellite CuS nanocrystals (NCs), causing metastable displacive ion movements. Electron diffraction measurements under photo illumination, ultrafast time-resolved electron diffraction analyses, and theoretical calculations of semiconductive plasmonic CuS NCs showed that metastable displacive ion movements due to the LSPR-induced cooperative Jahn–Teller effect delayed the relaxation of LSPR in the microsecond region. Furthermore, the displacive ion movements caused photo-switching of the conductivity in CuS NC films at room temperature (22 °C), such as in transparent variable resistance infrared sensors. This study pushes the limits of plasmonics from tentative control of collective oscillation to metastable crystal structure manipulation.

The science of localised surface plasmon resonance (LSPR), which was introduced by Faraday in 1857, is relevant in numerous scientific and industrial fields, including device applications, energy conversion, and catalysis[1–6]. LSPR involves the collective oscillation of free carriers and has been an important method to stimulate light–matter interactions by controlling the states of matter by changing properties and functions[7–11]. However, the light-to-matter interaction in LSPR of conventional noble metal nanocrystals (NCs) has been limited to the stimulation of the collective mode. Collective oscillation of free carriers in LSPR leads to stimulation of the phonon mode. The electron and phonon modes induced by LSPR are momentarily relaxed and less likely to influence the fundamental properties of the quantised material due to light. However, no exceptional relaxation processes have been found in the history of plasmonics. The causes of atom/ion displacement can have a great impact on the material over time. Although the

LSPR relaxation process, linked to the phonon mode, causes a thermal lattice expansion (also called the breathing/extensional mode)[12–14], there are no reports of metastable lattice rearrangement induced by LSPR. Therefore, achieving LSPR-induced stimulation of metastable atom/ion displacement, accompanied by dramatic changes in material properties, will push the limits of plasmonics.

Plasmonic semiconductors may exhibit properties different to those of conventional plasmonic metals because of their moderately discretised band structure[15–17]. Although CuS NCs (which are LSPR materials) have been extensively investigated[16–21], their relaxation mechanism remains unclear. The relaxation of CuS NCs has been assumed to be similar to that of metallic LSPR materials (such as noble metals); however, the effects on semi conductivity are significant. Ludwig et al. used X-ray transient absorption spectroscopy to report that, unlike metal NCs with a continuous band structure, CuS NCs

[1]Institute for Chemical Research, Kyoto University, Uji, Kyoto 611-0011, Japan. [2]Tsukuba Research Center for Energy Materials Science (TREMS), Faculty of Pure and Applied Sciences, University of Tsukuba, 1-1-1 Tennodai, Tsukuba 305-8573, Japan. [3]Fukui Institute for Fundamental Chemistry, Kyoto University, Sakyo-ku, Kyoto 606-8103, Japan. [4]Department of Molecular Engineering, Graduate School of Engineering, Kyoto University, Nishikyo-ku, Kyoto 615-8510, Japan. [5]National Institute for Materials Science (NIMS), 1-1 Namiki, Tsukuba, Ibaraki 305-0044, Japan. ✉e-mail: sakamoto@scl.kyoto-u.ac.jp; hada.masaki.fm@u.tsukuba.ac.jp

exhibit a carrier-trapping process peculiar to semiconductors in the ps region after LSPR excitation[21]. In addition to the general LSPR relaxation processes defining the consensus mentioned above, CuS NCs exhibit an inherently slow LSPR relaxation in the microsecond region (lifetime = 1.7 μs)[17]. This unique delay in relaxation is indicated by the laser flash photolysis results of CuS NCs, which confirm an extraordinarily slow LSPR-bleach recovery. However, the origin of slow-recovery of LSPR bleach has remained unclear because the spectroscopy data cannot directly track the ionic coordination in solids.

In this work, ultrafast time-resolved electron diffraction measurements with sub-picosecond time-resolution indicated that displacive ion movements by LSPR caused delayed LSPR relaxation of CuS NCs (Fig. 1). Theoretical calculations predicted an LSPR-induced cooperative Jahn−Teller (JT) effect. The LSPR-stimulated interaction between the localised orbital electronic state and crystal lattice induced a phase transition (i.e., cooperative JT effect)[22,23], allowing for ionic displacement in CuS NCs. The conductivity of a CuS NC film decreased upon LSPR excitation, following LSPR-induced JT distortion. Based on the obtained results, we demonstrated the infrared (IR)-responsive variable resistance of the CuS film using LSPR-induced JT distortion at room temperature. CuS NCs, which transmit visible light, have a transparent appearance. CuS NC transparent films can be applied to room-temperature-driven optical IR sensors owing to their advantages of transparency, fast response, and printability. The plasmonic control of metastable atomic configurations can expand the scope of plasmonics, facilitating an in-depth understanding of the fundamental relationship between the atomic configuration and essential properties of a substance, as well as its potential applications[24–27].

## Results

### Synthesis and characterisation of CuS NCs

CuS NCs with plate-like structures were synthesised according to a previous study[17]. Figure 2a shows transmission electron microscopy (TEM) images of CuS NCs (size: 16.3 ± 1.5 nm, thickness: 5.7 ± 1.1 nm). The X-ray diffraction (XRD) patterns show that the obtained NCs were composed of hexagonal covellite CuS (Inorganic Crystal Structure Database [ICSD] no. 26968) phases (Fig. S1).

The elementary cell of CuS exhibited hexagonal symmetry corresponding to a $P6_3/mmc$ space group with six formula units per unit cell (Fig. 2b)[28]. The atomic configuration of CuS consisted of three alternating layers (CuS4-tetrahedra layer/CuS3-triangles layer/CuS4-tetrahedra layer), where the CuS4-tetrahedra layers pinched the CuS3-triangles layers. In the CuS4-tetrahedra and CuS3-triangles, Cu ions exhibited tetrahedral coordination ($Cu_{Td}$) and trigonal coordination ($Cu_T$), respectively. The three sulphur atoms of a CuS3-triangle, with one atom shared with a CuS4-tetrahedron, were labelled S(1) (Fig. 2b), while the other sulphur atoms of the CuS4-tetrahedron were labelled S(2). The set of three alternating layers (CuS4-tetrahedra layer/CuS3-triangles layer/CuS4-tetrahedra layer) were connected along the $c$-axis via S(2)−S(2) bonds.

The calculated band structure of CuS is shown in Fig. 2c. At the Fermi level, the electronic states consisted of mixed valence states of copper. Holes were elastically scattered among the degenerate orbitals at the Fermi level, making them free from Cu-atom charges. This imparted a p-type semiconducting behaviour to CuS NCs, which consequently exhibited an LSPR peak at 1080 nm (Fig. 2d).

### LSPR-induced atomic displacement of CuS NCs

The electron-diffraction-pattern shift induced by light irradiation was directly observed to understand the LSPR relaxation process of a CuS NC from the viewpoint of the crystal structure (Fig. 3). Without light irradiation, the diffraction spots 1 3 0 and 2 3̄ 0 of the CuS NCs exhibited an almost similar intensity, with a slight difference caused by the tilting of the NC against the vertical of the incident electron beam. Under light irradiation, the 1 3̄ 0 diffraction spot exhibited a higher intensity than the 2 3̄ 0 spot, while the 1 3 0 spot exhibited a lower intensity than the 2 3 0 spot. The simulation of crystal structures exhibiting the diffraction-spot shift indicated a movement of $Cu_T$ (indicated by a red arrow in Fig. 3c; see Figs. S2−S4), which was validated by theoretical calculations (vide infra). The shift in the electron

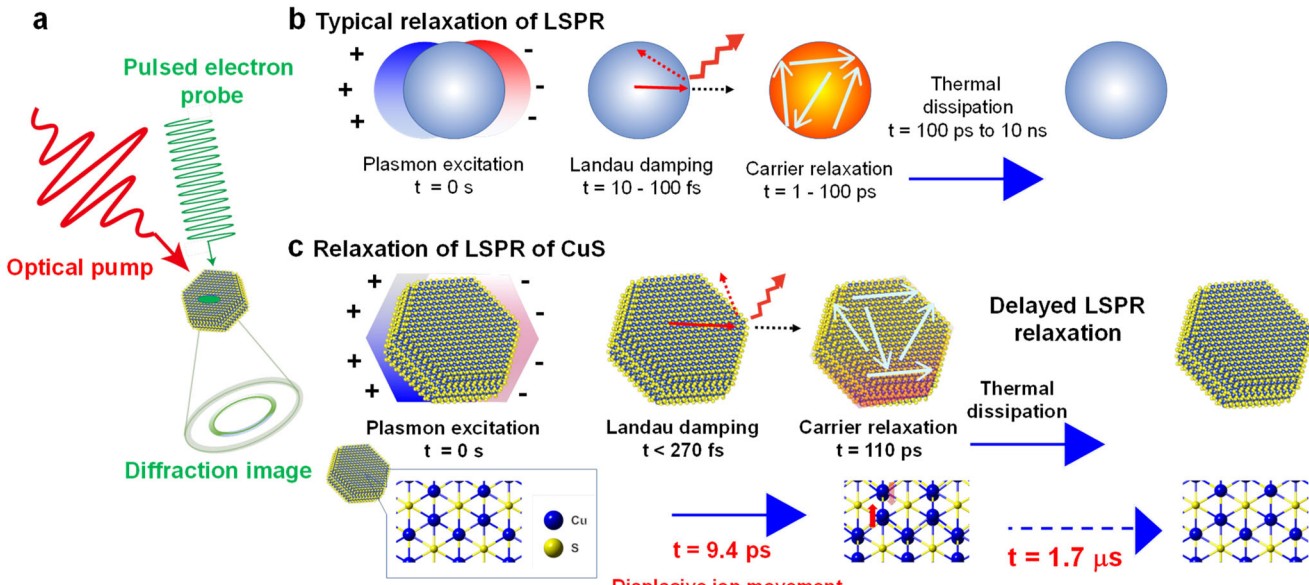

**Fig. 1 | LSPR relaxation of CuS NCs. a** Schematic representation of ultrafast electron diffraction used before optical-pump and electron diffraction probe experiments on colloidal nanocrystals (NCs) deposited on the substrate. **b** Typical relaxation process of LSPR[3]. LSPR involves the coherent oscillation of free carriers stimulated by light[3,45]. Its relaxation process proceeds through Landau damping, electron−electron scattering, electron−lattice scattering, and lattice−lattice scattering, corresponding to time domains of 10−100 fs, 1−100 ps, and 0.1−100 ns, respectively[3,46]. **c** LSPR relaxation process of CuS proceeds through Landau damping, carrier relaxation, and an unknown delayed relaxation corresponding to the time domains of <270 fs, 110 ps, and 1.7 μs, respectively[17]. The ionic displacements associated with LSPR excitation cause the delayed LSPR relaxation shown in the subsequent figures.

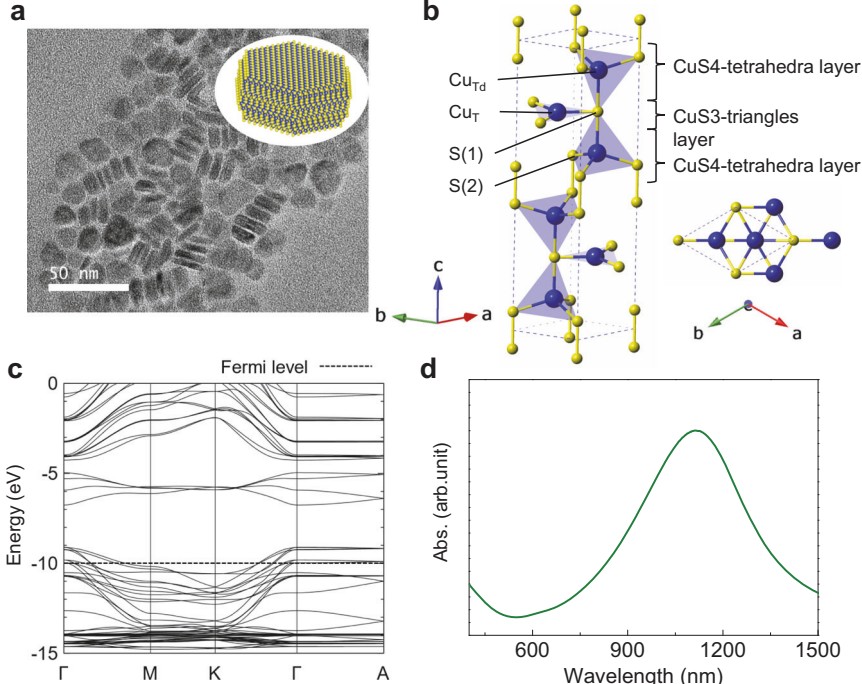

**Fig. 2 | Crystal structure and optical properties of the CuS NCs. a** TEM image of the CuS NCs. **b** Crystal unit cells of CuS with lattice constants of $a = b = 3.796$ and $c = 16.360$ Å. The yellow and blue spheres indicate sulphur and copper atoms, respectively. **c** Calculated band structure of CuS within the tight-binding approximation (see Method section for details). **d** UV–Vis–NIR absorption spectra of the CuS NCs in a chloroform dispersion. Abs absorption.

diffraction pattern indicated that the atomic displacement was a collective atomic rearrangement in the NCs.

The change in full width at half maximum of the diffraction spots could not be observed by light irradiation (Fig. S5), indicating that the long-range structural periodicity of the system was maintained during the light-stimulated ionic displacement.

**Time-resolved electron diffraction measurement**

Ultrafast time-resolved electron diffraction measurements were used to investigate the lattice relaxation of CuS NCs after LSPR excitation[24,26,29]. The electron diffraction pattern of the 60-nm-thick CuS NC film indicated two rings on the (012) and (110) planes, as shown in Fig. 4a, owing to the NCs being randomly oriented in the lateral direction and highly oriented (more than 70%) along the [001] axis (in the longitudinal direction). The 1 1 0 diffraction ring was more intense than the 0 1 2 diffraction ring (Fig. 4a); therefore, the peak intensities and positions (scattering vector: $Q = 1/d$, where $d$ is the lattice spacing) were analysed using the 1 1 0 diffraction ring.

The time evolution of the peak-position and intensity of the diffraction ring after irradiation with 400- and 800-nm light with an incident fluence of 5 mJ cm$^{-2}$ is shown in Fig. 4b and c, respectively. The laser with the fluence of 5 mJ cm$^{-2}$ was stable and almost constant (error bars: 800 nm: ~1%, 400 nm: ~2%). The 400- and 800-nm light-induced bandgap excitation and LSPR excitation, respectively, are also shown in these figures. Below an incident fluence of approximately 8 mJ cm$^{-2}$, the photoexcitation phenomenon occurred in a repetitive regime. Light excitation (via 400- and 800-nm light) caused a negative shift of the $Q$-value of the (110) plane (Fig. 4b), indicating a lattice expansion of the CuS NCs by laser heating. Owing to the size effect of nanocrystals, the CuS lattice could expand on a slight temperature rise. As shown in Fig. 4c, excitation of the LSPR band using an 800-nm laser changed the intensity of the 1 1 0 diffraction ring; this differed from bandgap excitation. The intensity of the 1 1 0 diffraction ring decreased by approximately 6% with a 9.4 ± 3.1 ps excitation of the LSPR band. Almost half of the intensity shift (~3%) was recovered, with a time

constant of 45 ± 16 ps, while the other half remained low, and no temporal shift was observed within the time window provided by the instrument used. This could indicate a large lattice expansion immediately after LSPR excitation, subsequently transforming into a metastable crystal structure. This result agrees with the TEM observation of a shift in the diffraction spot under light irradiation.

Intensity changes in diffraction rings are caused by atomic movements in a unit cell or the Debye–Waller effect[30]. As shown in the Supplementary Information (Figures S6 and S7), laser heating with 800-nm light caused a temperature increase of 11 K. The modulation in diffraction intensity induced by this is less than 1%[30]. Thus, the photo-induced intensity changes in the diffraction pattern of the CuS NCs could not be attributed to the simple photo-thermal Debye–Waller effect. The decrease in diffraction intensity, which was not observed under bandgap excitation (under 400-nm light), indicated that LSPR excitation (under 800-nm light) induced ionic displacements in the unit cell without a change in the long-range structural periodicity (Fig. 4c).

Diffraction intensity shifts reflect local or cooperative ionic displacements; thus, the temporal shift of the half-width ($2\sigma$) of the diffraction peak was investigated. As an inhomogeneous distortion in NCs induces width broadening, a local ionic displacement should increase width. However, as shown in Fig. 4d, the diffraction width did not shift significantly after LSPR excitation, indicating that the observed ionic displacement was a cooperative phenomenon. Thus, LSPR excitation changed the crystal structure of the NCs to a metastable state.

We investigated the time evolution of the peak-position and intensity of the diffraction ring after excitation with 1053- and 527-nm laser light using a Q-switched nanosecond laser for an irradiation duration of 10 ns (Supplementary Information Fig. S8). 1053- and 527-nm laser excitation caused a negative shift in the $Q$-value of the (110) plane (Figs. S7a and c), indicating that the CuS NCs underwent lattice expansion owing to laser heating. Diffraction intensity decreased under LSPR excitation (1053-nm light) but not bandgap excitation (527-nm light), indicating that LSPR excitation induced ionic

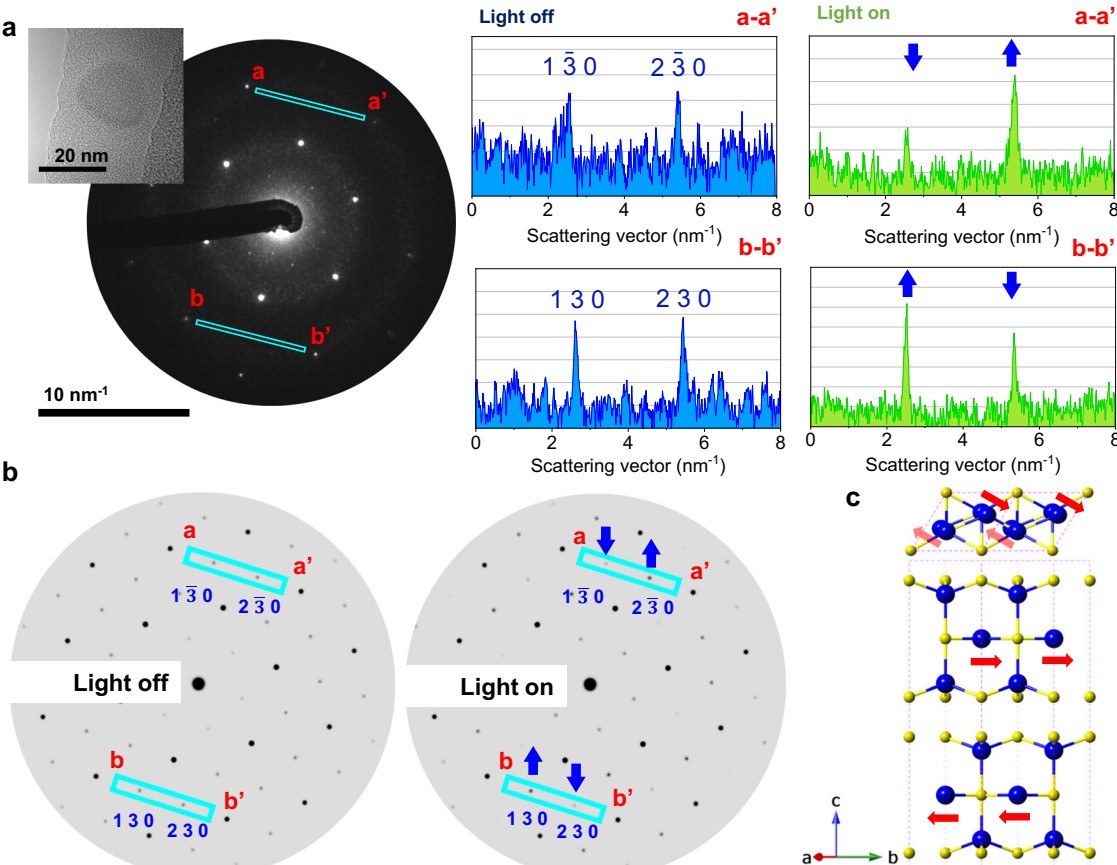

**Fig. 3 | Light-stimulated shift of the CuS NCs electron diffraction and its simulation. a** Electron diffraction pattern of CuS NC under continuous-wave light irradiation from Xe lamp with an intensity of approximately 140 mW/cm² (Cermax PE300BF). Inset: TEM image of CuS NCs corresponding to the electron diffraction pattern. The intensities of the diffraction peak in the blue rectangular region with and without light irradiation are shown on the right-hand side. The blue arrow indicates the intensity shift of the coupled diffraction spots (1 $\bar{3}$ 0 against 2 $\bar{3}$ 0 and 1 3 0 against 2 3 0) under light irradiation. **b** Simulated electron diffraction patterns of CuS NC with and without light irradiation. **c** $Cu_T$ shifts causing an electron diffraction pattern corresponding to the ionic motion indicated by the red arrows (see the Supplementary Information for details).

displacements in the unit cell (Figs. S8b and d). These results support that the shift in the crystal structure of the NCs stimulated by the LSPR excitation.

A threshold-like structure for the laser fluence-dependent shift of electron diffraction intensity from the (110) planes (Supplementary Information Fig. S9) was observed. The power dependence of continuous-wave (CW) IR light (800 nm)-responsive photoconductivity of the CuS NC film shows a similar tendency (*vide infra*). The LSPR-driven phenomena are induced both by CW and pulsed light.

To better understand the influence of surface effects such as multiphoton ionisation and thermalisation on the reduction of electron diffraction intensity, we investigated the effects of the pulse duration of the fs-laser on the time evolution of the electron diffraction intensity from the (110) planes (Fig. S10) and image of the photoemission from the sample surface (Fig. S11). If the surface effects contribute to the IR-photoinduced phenomena in CuS NCs, the pulse width significantly affects the time evolution of electron diffraction intensity and surface image of photoemission. However, no significant shift was observed due to varying pulse durations. This result supports the hypothesis that multiphoton phenomena does not contribute to the current IR-photoinduced phenomena in CuS NCs.

To further discuss the LSPR-induced ionic displacement in CuS NCs, the time evolution of the intensity and peak position of the diffraction ring immediately after LSPR excitation (Fig. 4e and f) were investigated. A shift in the $Q$-value occurred within the pulse duration of the probe electron pulse (<1 ps), and the time constant indicated

that this rapid shift corresponded to the laser heating-mediated lattice expansion observed in LSPR materials[13,14]. In contrast to the $Q$-value shift, the intensity of the 1 1 0 diffraction ring gradually shifted after laser excitation, exhibiting a time constant of $9.4 \pm 3.1$ ps. Subsequently, the shift was recovered with a time constant of $45 \pm 16$ ps, indicating stabilisation to a metastable state. The time constant indicated that the displacive ion movements occurred in the carrier- or carrier-phonon-scattering stage during LSPR relaxation.

## Theoretical calculations

Theoretical calculations within the tight-binding approximation (see Methods section for details) were used to qualitatively analyse the electronic structures of CuS that caused the experimentally observed $Cu_T$ movement under 800-nm light excitation. The time-resolved electron diffraction measurements under light irradiation showed that the ions moved in the same direction in each unit cell (Fig. 3c), suggesting that the 800-nm light caused electronic transitions at the Γ point. An enlarged view of the band structure around the Γ point (Figure S12a) showed that the lowest unoccupied crystal orbital (LUCO) is degenerated at the Γ point. The electron excitation to the degenerate LUCO from the lower-energy orbitals at the Γ point induced the JT effect. One of the degenerate LUCOs has bonding characteristics and the other anti-bonding characteristics between the $Cu_T$–S(1) bond (Figure S12b). Therefore, the excitation of the LUCOs generated ionic displacements, shortening the $Cu_T$–S(1) bond. The planar triangle consisting of $Cu_T$ and three S(1), where the LUCOs

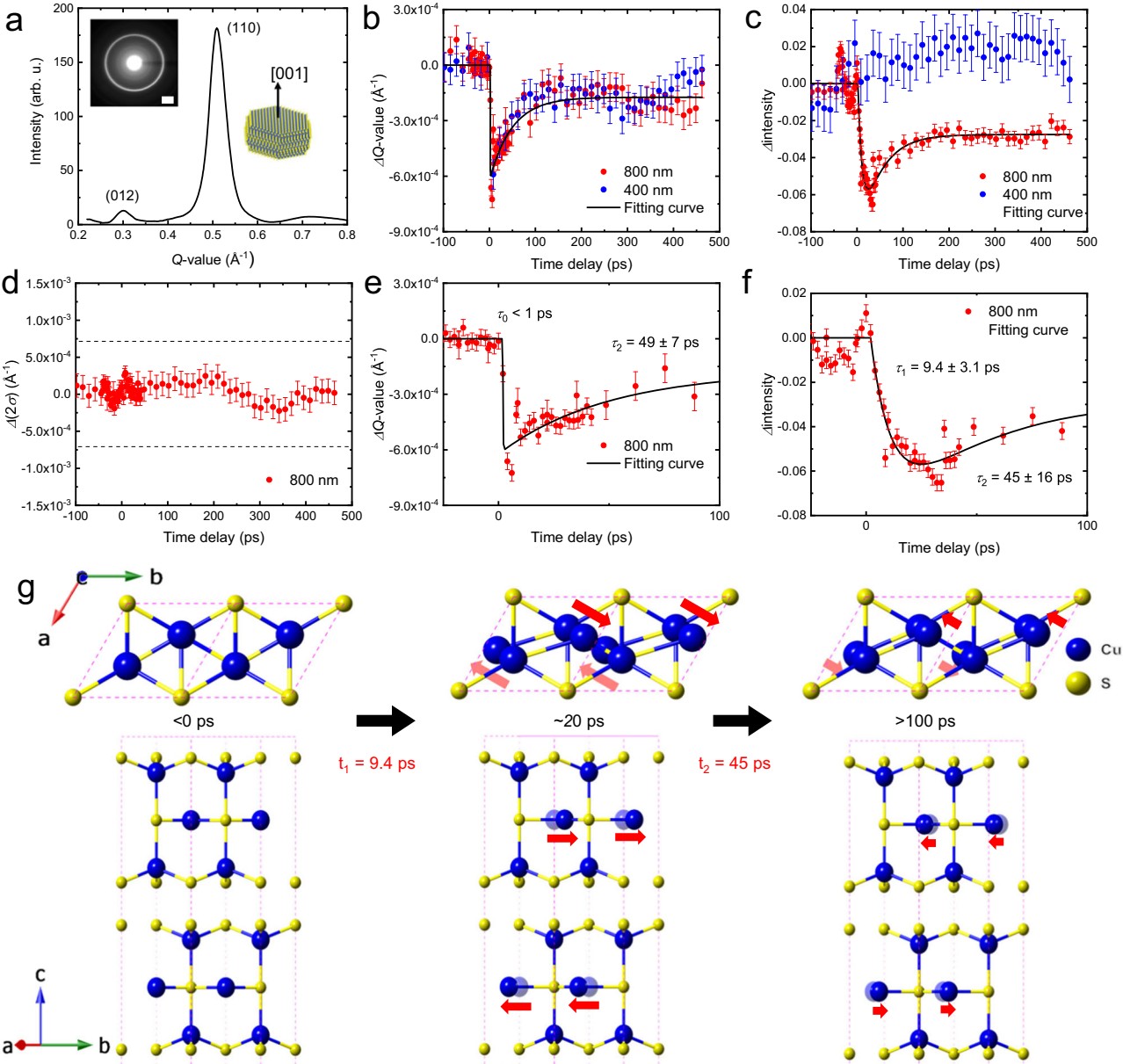

**Fig. 4 | Time-resolved electron diffraction of the CuS NCs. a** Radially-integrated diffraction intensity of the CuS NCs film. Inset: Electron diffraction pattern. The white scale bar represents 0.2 Å⁻¹. **b** Time evolution of the $Q$-value from the (110) planes under fs-laser excitation in the wavelength range of 400 (blue circle)–800 (red circle) nm. The error bars represent the standard deviation at each time delay. **c** Time evolution of the electron diffraction intensity from the (110) planes under fs-laser excitation in the wavelength range of 400 (blue circle)–800 (red circle) nm. The error bars represent the standard deviation at each time delay. **d** Temporal shift in the width of the electron diffraction intensity from the (110) planes under fs-laser excitation at 800 nm (red circle). The dotted blue lines indicate the peak shifts derived from (**b**). The error bars represent the standard deviation at each time delay. **e** Magnified time evolution of the $Q$-value obtained from the (110) electron diffraction rings under fs-laser excitation at 800 nm (red circle). The solid black line indicates the best fit. The error bars represent the standard deviation at each time delay. **f** Magnified time evolution of the electron diffraction intensity from the (110) planes under fs-laser excitation at 800 nm (red circle). The solid black line indicates the best fit. The error bars represent the standard deviation at each time delay. **g** Illustration of the light-stimulated ionic displacement occurring in CuS NCs.

are distributed, has $D_{3h}$ site symmetry. Thus, the irreducible representation of the degenerate excited state localised around this site is considered to be $E'$ or $E''$. Their symmetrical product is $[E'^2] = [E''^2] = A_1' \oplus E'$, indicating that the degenerate excited electronic state couples with a vibrational mode (with representation $e'$) to give the $E \otimes e$ problem. The epikernel of $e'$ in the parent group $D_{3h}$ is $C_{2v}$:[31]

$$E(D_{3h}, e') = C_{2v} \qquad (1)$$

Thus, the excitation of the degenerate LUCO at the Γ point induced the $E \otimes e$ JT effect, lowering the site symmetry around $Cu_T$ from $D_{3h}$ to $C_{2v}$ with shortening the $Cu_T$–S(1) bond. These results agreed with ion movements predicted by the time-resolved electron diffraction measurements. During the shift of the 1 1 0 diffraction ring upon LSPR excitation (approximately 7%), the $Cu_T$ moved approximately 0.19 Å (9.5 ps after LSPR excitation). Subsequently, the ionic-displacement shift of $Cu_T$ settled into a metastable state with a 0.165 Å $Cu_T$ shift and a time constant of 45 ps.

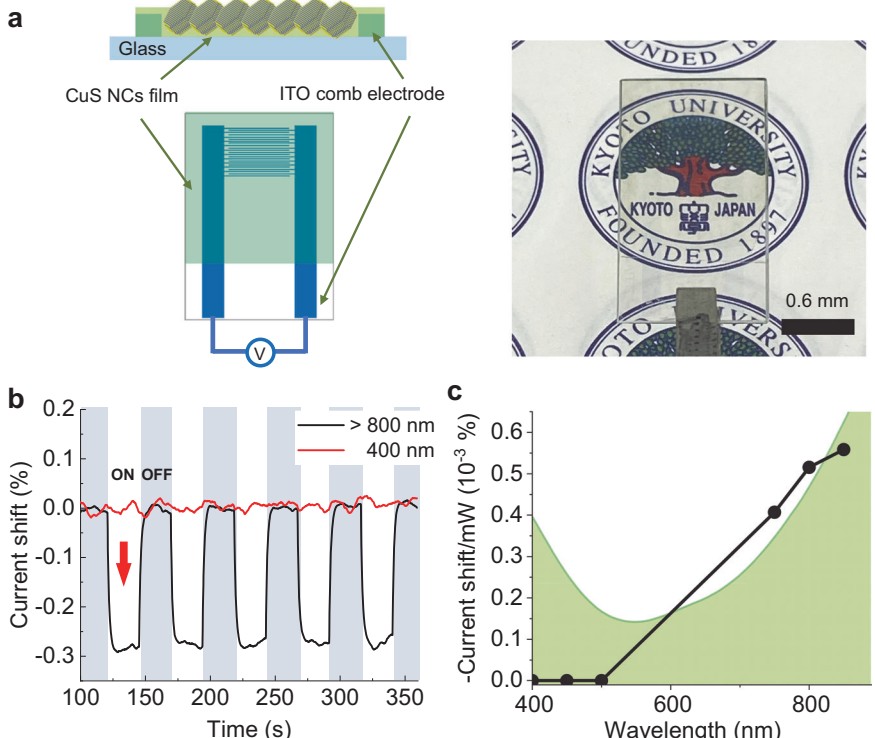

**Fig. 5 | Device application of the light-stimulated ionic displacement in CuS NCs. a** Illustrations of devices (left-hand image). Photograph of a CuS NCs film coated on an ITO comb electrode (right-hand image). See also Figure S14 for details. **b** IR light ( > 800 nm)-responsive photoconductivity (central image) of the CuS NC film at a voltage of +1 V. The power dependence of photoconductivity shift was shown in Figure S14. **c** Excitation wavelength dependent on photoconductivity shift of the CuS NC film divided by the power of light at a voltage of +1 V. The green line indicates the extinction spectra of CuS, indicating that a shift in resistance occurs in response to LSPR.

## Device application

To analyse the LSPR-induced displacive ion movement as a macroscopic phenomenon, the LSPR-responsive photoconductivity of the CuS NCs film was investigated (Fig. 5). A thin film of CuS NCs (thickness: 50 nm) was prepared on a comb electrode using the layer-by-layer deposition method (see Method section for details). The CuS NCs film was set on a temperature controller with a heat sink and a Peltier element to avoid any thermal effect of infrared irradiation on its conductivity. IR-light irradiation caused a decrease in its conductivity (Fig. 5b). The CuS film showed semiconductive behaviour in temperature-dependent conductivity measurements; thus, an increase in temperature via IR light should not decrease the conductivity of the film (see Figs. S13 and S14). Additionally, a shift in the conductivity ratio against photon flux reproduced the LSPR peak of CuS (Fig. 5c). Furthermore, the threshold for the fluence dependence of conductivity reflected the threshold of laser fluence for ionic displacement observed in the time-resolved electron diffraction (Fig. S15). Therefore, the decrease in conductivity by IR irradiation could be attributed to the change in crystal structure induced by LSPR. The LSPR of CuS NCs underwent a redshift under IR-light irradiation[14], indicating that the number of free carriers contributing to the conductivity decreased under IR-light irradiation. The shift of LSPR validated this result under IR-light irradiation[14]. The CuS transparent film would thus be applicable to room-temperature-driven optical IR sensors. The advantages of transparency, fast response, and printability of the CuS films are favourable for invisible wearable IR sensing devices and/or neural sensing, among others[32,33].

## Discussion

The entire LSPR relaxation process of CuS NCs is chronologically shown in Scheme 1c. LSPR excitation stimulated lattice expansion by laser heating within 1 ps, which induced the phonon mode (correlated to the expansion coordinate of the NC)[34] and caused inter-band transitions, stimulating the JT mode. The stimulated JT mode caused displacive ionic movements of $Cu_T$ at approximately 10 ps. The strain induced by the cooperative JT effect induced an LSPR redshift and decrease in conductivity. The twist in the crystal structure (i.e., atomic displacement) was released after eliminating the JT distortion, indicated by the delayed relaxation of LSPR.

Lindenberg et al. reported that localised carrier trapping, which forms the surface polarons in quantum dots (QDs)[35], also induced lattice distortion. Krawcyzk et al. further investigated the mechanism in PbS QDs and suggested that the multiphoton absorption carrier-trapping process stimulates the lattice distortion in a QD[36]. However, the phenomenon exhibited in this study was not caused by carrier-trapping distortion but by the cooperative JT distortion of the NCs. Lattice distortion in a semiconductor QD by bandgap excitation is localised and not cooperative[35,36], whereas LSPR-induced hot-carrier generation is comprehensive, as indicated by the width shift and electron diffraction under light irradiation.

The LSPR-induced cooperative JT effect described herein enabled a crystal structure to shift with an extremely long lifetime due to the unique nature of the effect. The material design based on vibronic coupling theory could be applied to the development of drastic and long-lived LSPR-induced ionic displacement with desired functional switching[37]. The combination of LSPR and the cooperative JT effect could guide the atomic-configuration manipulation of plasmonic nanomaterials in future research. The observations in this study could also be applied to light stimulation of physical phenomena influenced by the cooperative JT effect, such as metal–insulator transition, superconductivity, and so on[7,38,39]. Furthermore, the LSPR-induced JT effect could provide an alternative to

elucidate the interaction between functions and atomic configuration in NCs.

## Methods

### Synthesis of CuS NCs

Hexagonal plate-like CuS NCs were synthesised according to a previous study[14]. A mixture of copper (I) acetate (0.123 g, 1 mmol) and oleylamine (10 mL) was degassed at 160 °C for 30 min. Subsequently, a solution of sulphur (0.048 g, 1.5 mmol) in 1-octadecene (15 mL) was injected rapidly into the mixture under a nitrogen atmosphere and stirred for 10 min. The resulting product was purified by adding an ethanol-hexane ($v$:$v$ = 1:1) mixed solvent to the solution, centrifuging twice, and then redispersing the precipitate in hexane.

### Characterisation

High-resolution TEM (HRTEM) characterisation was carried out using a JEM-2200FS (JEOL) electron microscope at an acceleration voltage of 200 kV. XRD patterns were recorded on a PANalytical Aeris diffractometer, with Cu Kα radiation ($\lambda$ = 1.542 Å) at 40 kV and 15 mA. Ultraviolet–visible–near-infrared (UV–Vis–NIR) absorption spectra were recorded using a UV-3600 spectrophotometer (Shimadzu).

### Time-resolved electron diffraction measurements

Ultrafast time-resolved electron diffraction was performed on CuS film (60–70 nm) on the ultrathin silicon nitride substrate under a vacuum condition. Femtosecond laser with a central wavelength of 800 nm, a repetition rate of 1 kHz, and a pulse duration of 110 fs was used for ultrafast time-resolved electron diffraction. On the probe arm, a UV pulse (wavelength: 267 nm), converted from NIR pulse (wavelength: 800 nm), produced ultrashort (~1 ps) electron pulses via the photoelectron effect[30]. A 75-keV electrostatic field accelerated the electron beam. The CuS film was excited by near-UV and NIR pump pulses (wavelengths of 400 and 800 nm, respectively) synchronised with electron pulses. The pulse durations of pump pulses were approximately 110 fs. The fluency of the exciting pulses was set to 1.5–5 mJ cm$^{-2}$. The electron beam diffracted by the CuS film under photoexcitation was detected by a charge-coupled device camera. For nanosecond-time-resolved electron diffraction measurement in the Supplementary Information, we used a Q-switched nanosecond laser with an irradiation duration of 10 ns.

### Simulation of LSPR-induced atomic displacement

Electron diffraction patterns were calculated using the CrystalMaker®, CrystalDiffract®, and SingleCrystal® software packages, based on kinematic theory. Covellite CuS, with the $P6_3/mmc$ space group, was used as the initial structure during analysis. The crystal structure and NC in the figures were prepared using VESTA®[40]. The atomic motions in CuS induced by photoexcitation were derived using theoretical calculations.

### Theoretical calculations

The electronic structures of CuS were computed using the extended Hückel theory[41] with the parameters provided in YAeHMOP[42]. The Brillouin zone was sampled with 405 $k$-points selected based on the method of Ramirez and Böhm[43]. The extended Hückel theory was employed instead of the density functional theory (DFT). This is because the DFT calculations of CuS resulted in a zero-energy gap near the Fermi level[44], although the 400-nm light is known to induce the bandgap excitation experimentally. The DFT + $U$ calculations also yielded a significantly small energy gap (-0.2 eV) in comparison with the 400-nm light[44]. These indicate that DFT cannot reproduce the experimental observation. Moreover, the extended Hückel theory describes the band structure well, consistent with the experiments.

### Conductivity measurement of the CuS NCs film

A CuS NCs film with a thickness of 60 nm was fabricated on Au and ITO comb (GEOMATEC Co., Ltd.) electrodes using layer-by-layer deposition of octan solutions of CuS NCs and short ligands (0.3 % KSCN solution of methanol) via ligand exchange. A 300 W Xe lamp (Cermax, Excelitas Technology) equipped with a UV–Vis cutoff filter ($\lambda$ > 800 nm, power density: 50 mW cm$^{-2}$) was used as the light irradiation source. To avoid any thermal effect on conductivity, the electrode was set on a Peltier device during IR-light irradiation.

## Data availability

The data generated in this study are provided in the Supplementary Information and Source Data file, or from the corresponding author upon request. Source data are provided in this paper.

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

## Acknowledgements

The temperature-dependent XRD experiments were conducted at the BL5S2 of Aichi Synchrotron Radiation Center, Aichi Science & Technology Foundation, Aichi, Japan (Proposal No. 202206170). M.S. is grateful to Tetsuri Nishikawa, Hiroki Morishita, and Norikazu Mizuochi for Hall measurements, Keito Sano for XRD measurement, Hsu Shih-chen for sample preparation and M.H. is grateful to Yuri Saida, Yui Iwasaki, Shin Ueno, and Wataru Yajima for experimental assistance. M.S. and M.H. are grateful to Satoshi Ohmura for fruitful discussions. F.U. is grateful to Dr. Ayako Hashimoto at NIMS for supporting the light-irradiation TEM experiment. This work was supported by the 'Joint Usage/Research Program on Zero-Emission Energy Research' of the Institute of Advanced Energy, Kyoto University (ZE2020C-8), NIMS Electron Microscopy Analysis Station, Nanostructural Characterisation Group, and JSPS KAKENHI Grant Numbers: JP22K05253 in Scientific Research (C) (T.S.), JP21H04638 in Scientific Research (A) (M.S.), JP20H01832 in Scientific Research (B) (M.H.), and JP20H04657 in Scientific Research on Innovative Areas (M.H.). This work was also supported by the JST FOREST Program (Grant Number JPMJFR201M, JPMJFR211V) (M.S., M.H.), the Adaptable and Seamless Technology transfer Program through Target-driven R&D (A-STEP): JST Grant Number JPMJTR20T1 (M.S.), and the Ministry of Education, Culture, Sports, Science and Technology (MEXT) Program for Development of Environmental Technology using Nanotechnology (Global Research Center for Environment and Energy Based on Nanomaterials Science) (F.U.). A part of this work was supported by NIMS microstructural characterization platform as a program of 'Nanotechnology Platform' of MEXT Grant Number: JPMXP09A21NM0083 (F.U.). Numerical calculations were carried out using the Supercomputer System of the Institute for Chemical Research, Kyoto University, Academic Center for Computing and Media Studies (ACCMS), Kyoto University, and the Research Centre for Computational Science, Okazaki (Project: 22-IMS-C065).

## Author contributions

M.S. conceived the concept of this work and designed the all experiments and carried out material fabrication and device application. M.H. carried out the time-resolved electron diffraction measurement, and shadow imaging experiments. F.U. carried out the electron diffraction measurement in TEM with and without light irradiation and simulation of atomic displacement. T.S. and W.O. carried out the theoretical calculations. M.S. wrote the manuscript. All authors participated in the discussion of the research.

## Competing interests

The authors declare no competing interests.
