## [Peer Review File · Nature Communications]

REVIEWER COMMENTS

Reviewer #1 (Remarks to the Author):

The concept of a localized surface plasmon coupling specifically to ionic motions to induce structure changes, within the language of Jahn-Teller distortions, is interesting and worthy of publication in Nature Communication. The multiple different experiments and theory provide a fairly consistent picture of a very unusual mechanism leading to lattice displacements and control of semiconducting/optical properties of the CuS nanoparticle films. The work is from a very strong team of which I have no doubt of the care taken in the execution of the experiments and theory.

There are a few points, however, that need to be addresses so that the authors do not assign the interesting effect to a mechanism that is not the underlying physical cause. The observations and overall applications are not affected but some clarity of the operating photophysics may be needed.

First, minor points:

- It is stated "Details of the ultrafast time-resolved electron diffraction setup are mentioned in another publication.²²". This reference is actually to Hartland's work. Similarly, there is an error in references to the work of Lindeberg et al "Lindenberg et al. have reported that localised carrier trapping, which forms the surface polarons in quantum dots (QDs),²⁶" Which I believe should be ref. 37.

- in regards to ref. 37, the authors should cite the work of Krawczyk et al "Nonthermal Lattice Disorder Observed in Photoexcited PbS Quantum Dots" J. Phys. Chem. C 2021, 125, 40 (2021). This latter reference is important to include as it points out the problem with ref. 37 (Lindeberg et al) in that the effect they observe is likely multiphoton ionization that has a much higher cross section at 400 nm than the band gap excitation. This point is important as they neglect multiphoton absorption and have to propose as surface hole state fully resonant with the band states. There is no way this would lead to a long lived trapped hole". I make this point as the authors need to think about their excitation levels.

- Figure 3a. I could not understand the point of the photo. The authors should show where the field is applied and affect on transmission better. At least I could not understand to what effect other than a uniform filter the effect gave that would not be unusual. Please improve this figure to make the application more apparent.

- Please remove reference Faraday's in the abstract namely "thereby expanding on Faraday's discovery". It is unnecessary and gives the impression that there was some unknown physics that is now solved by this work. The historical reference in the introduction is fine and makes the intended point.

Major concern. I only really have one and that is that all of the noted effects could be due to changes in T with the very high peak powers used. The characterization and use of optical absorptivities all assumes weak perturbation in the 1-photon limit. However, the short excitation pulses used puts the excitation levels at 50 GW/cm² assuming 100 fs pulses. (Please give more details on the excitation pulses and whether 5 mj/cm² was kept constant for 400 nm and 800 nm with error bars). There is no doubt in this reviewer's mind that there is multiphoton absorption occurring at these peak powers. The estimated T jump is far too low an estimate. I am not even sure how a local surface plasmon could have such a low absorptivity even in the weak perturbation limit.

The authors must measure the absorbed excitation at the different fluences used, and do a power dependence. This point is made specifically in light of the observation that above 8 mj/cm² the excitation is no longer reversible. If the delta T changes are as small as the authors estimate, then this onset of damage at 8 mj/cm² makes no sense. Surely if they heat the films up by 11 degrees to 30 degrees above RT, they would not see the films change such that if their delta T estimates were correct the process should be completely reversible to much higher excitation, especially given this delta T is for the absorbed laser energy prior to diffusion. The actual delta T under steady state KHz excitation in data collection would be much less and should be completely reversible.

I will reserve judgement until this control is done but I suspect that the note effect is thermal in origin. At these excitation peak powers and strong electron-electron coupling in the 800 nm with respect to plasmon formation that the system is close to the formation of lattice breakdown or preplasma from many sources of multiphoton emitted electron. The absorption skin depth would then be more comparable to preplasma formation in nominally transparent materials at higher excitation. The change in lattice T could be much higher and the effect of resonant multiphoton ionization would also affect the lattice stability. The dual effect of fully resonant multiphoton ionization, collective coupling to the surface plasmon, could lead to very large T jumps sufficient to drive lattice displacements for the now highly perturbed lattice.

I want to assure the authors I strongly support publication of this work in Nature Comm. I think I would not be doing my job if I did not ask for a power dependence on the time resolve diffraction data as well as transient absorption spectra. The absorption of the excitation pulse must be measured directly and not use calculated values based on weak perturbation optical properties.

This manuscript represents a beautiful set of experiments with clearly a very large lattice change with ionic motions strongly modulating the CuS nanoparticle properties. I fully support publication and hope the authors find the above suggestions helpful.

Reviewer #2 (Remarks to the Author):

see separate sheet

Reviewer #3 (Remarks to the Author):

In this manuscript, it is confirmed that localised surface plasmon resonance (LSPR) induced the cooperative Jahn Teller effect in covellite CuS nanocrystals (NCs), causing metastable displacive ion movements with several structural analysis and pump-probe dynamics and supported with suitable calculations.

Based on the existing studies, this research has been well conducted in a groundbreaking way. Moreover, there are attempts to overcome the limitations of the existing studies such as failure to track the ionic coordination or elucidate the mechanism caused by LSPR.

However, additional explanations and experiments to reinforce the arguments of the manuscript are still needed.

See attached file

The present manuscript discusses an interesting and timely topic, a plasmon-resonance induced Jahn-Teller distortion. The authors report on an interesting series of experiments performing state of the art investigations. Upon exciting a film of CuS nanoparticles, a different response is reported depending upon the laser wavelength. A pronounced change of optical properties is observed for an excitation below (!) the band gap. This is attributed to a change in atomic arrangement. Subsequently, the authors characterize this change in atomic arrangement and the time scale of the return to the initial state. Finally, the authors demonstrate a potential application by showing a photoconductivity change. This is an interesting sequence of experimental data and I hence recommend acceptance of this manuscript after making several improvements.

These comments and suggestions are listed in the following chronologically:

- a) I found the first two paragraphs of the main text quite vague. For example, what are the limits of plasmonics? How will they be pushed by the present investigation? Which elements of what relaxation mechanism are unclear?
- b) On p.5, the authors should say that their NCs are platelets, then the numbers for the lateral diameter and thickness are easier to understand.
- c) The authors might want to replace the phrase 'global phenomenon' by 'collective atomic rearrangement'
- d) The caption of figure 2 does not say anything about the laser pulse needed (pulse length and power).
- e) It is confusing that the label: figure 2, is shown twice. Place label all figures sequentially from 1 to 4.
- f) The authors argue on p. 11 that the decrease of diffraction intensity was not shown for bandgap excitation, but their data in figure 2.b show a similar change in atomic rearrangement. These explain this apparent contradiction.
- g) In figure 3.b (should be 4.b), the caption does not specific the light intensity needed to show the effect. It also would be nice to know if the effect depends upon the light intensity in a linear fashion.
- h) The authors argue that the crystal structure shift is extremely long. Yet, this shift only lasts for 50 ps. Why is this extremely long? Please explain this in more detail.

authors have responded in sufficient detail to the questions, comments and criticism of the two reviewers. This is clearly appreciated. Hence, I suggest to accept the manuscript.

At the same time, I would like to summarize where my personal opinion and belief is not compatible with the models and explanations suggested in the present manuscript. I leave it up to the authors completely, if they respond to these comments or prefer to ignore them.

- a) Tetrahedral Ge:
Indeed, many groups have adapted initially adopted the tetrahedral Ge model by Tominaga, Kolobov and coworkers to explain the atomic arrangement in amorphous phase change materials like GeTe or Ge₂Sb₂Te₅. There is no doubt that this model can account for a number of experimental observations, such as the pronounced property contrast between the amorphous and the crystalline state. Yet, subsequent studies have shown both experimentally and theoretically that another model, where the amorphous material is characterized also by an octahedral atomic arrangement (like the crystal) provides a better explanation of the properties of the amorphous system. Aging is one of these phenomena. It appears as if amorphous GeTe 'ages' and upon doing so develops properties which are even more dissimilar from the crystalline phase. This is explained easily if one assumes an

increasing Peierls distortion in the amorphous phase, in line with the observed changes of the optical properties. Simulations show rather unequivocally that in the amorphous phase upon relaxation the number of tetrahedral sites decreases and the distortion of the octahedral sites increases.

The authors seem to indicate that tetrahedral Ge is stabilized in Te rich compounds such as GeTe₂. This is indeed a possible scenario. Yet, under Te-rich conditions it seems questionable if an octahedral arrangement would be a plausible scenario as the lowest energy state.

Finally, the Stanford group (E. Pop and coworkers) that recently also reported on superior switching in Sb₂Te₃ / GST₂₂₅ superlattices and commented that this improvement is only achieved for very thin films of Sb₂Te₃, implying that it is the finite thickness of the Sb₂Te₃ layer which is crucial for the superior behavior of these superlattices [*Nano Lett.* 2022, 22, 15, 6285–6291]. This observation cannot easily be explained by the tetrahedral Ge model discussed here.

b) Sb₂Te₃ (2D)

The authors seem to argue that Sb₂Te₃ is a 2D material. Sb₂Te₃ is clearly more anisotropic than GeTe and has a layered structure. Yet, the interaction across the Te – Te gaps is much larger than for typical layered materials like graphite or hexagonal BN. Hence, the label as a 2D material seems misleading. In this context the authors also compare chemisorption vs. physisorption. The concept of chemisorption would not be compatible with true 2D behavior. Also, recent publications have shown that the layer spacing in Sb₂Te₃ is too close compared with true van der Waals bonded systems, which show a larger distance across the gap.

Finally, I just note in passing that the phrase *inherent vacancy* might be replaced by the word *'stoichiometric vacancy'*. In GeSb₂Te₄, this is one stoichiometric vacancy within each unit cell, nevertheless the material has a band gap and is a semiconductor. *Excess vacancies* would make this material metallic, if sufficiently high numbers of excess vacancies exist.

The authors might want to consider these comments, but I do not request any further change of their manuscript.

In this manuscript, it is confirmed that localised surface plasmon resonance (LSPR) induced the cooperative Jahn Teller effect in covellite CuS nanocrystals (NCs), causing metastable displacive ion movements with structural analysis and pump-probe dynamics and supported with suitable calculation. However, some revisions on manuscript are needed. Questions about the manuscript are as follows:

Comments.

1. The reason why the plasmon excitation and relaxation processes of CuS, which is not a noble metal, are all on the ps scale, but the combined time scale is us scale is needed.
2. In the process of claiming that the ionic displacement observed in Figure 3.d is a cooperative phenomenon, the diffraction width was arbitrarily designated and it was stated that the width did not change. Additional evidence is needed to support the above claim.
3. It is stated that LSPR excitation in CuS did not change the long-range structural periodicity of the system. Is there any data such as SAED pattern or X-ray diffraction which can support the above invariability?
4. In this study, laser excitation and photoconductivity measurements were performed with 400nm (which is correspond to band transition region) and the 800nm wavelength (which is correspond to LSPR region) to show that the cooperative Jahn-Teller effect is due to LSPR. Are there other data sets where the same experimental splits were performed with the wavelength value of the 600nm (middle regime) or 1000nm (high value in LSPR regime)?

A point-by-point response to the reviewers' comments for NCOMMS-22-46751-T

REVIEWER COMMENTS

Reviewer #1 (Remarks to the Author):

The concept of a localized surface plasmon coupling specifically to ionic motions to induce structure changes, within the language of Jahn-Teller distortions, is interesting and worthy of publication in Nature Communication. The multiple different experiments and theory provide a fairly consistent picture of a very unusual mechanism leading to lattice displacements and control of semiconducting/optical properties of the CuS nanoparticle films. The work is from a very strong team of which I have no doubt of the care taken in the execution of the experiments and theory.

There are a few points, however, that need to be addresses so that the authors do not assign the interesting effect to a mechanism that is not the underlying physical cause. The observations and overall applications are not affected but some clarity of the operating photophysics may be needed.

Response: Thank you for the valuable comments, suggestions, and insights as well as the positive feedback. We have addressed all the comments individually. We hope that the explanation provided and revisions made in our work are clear and sufficient.

First, minor points:

- It is stated "Details of the ultrafast time-resolved electron diffraction setup are mentioned in another publication.²²". This reference is actually to Hartland's work. Similarly, there is an error in references to the work of Lindeberg et al "Lindenberget al. have reported that localised carrier trapping, which forms the surface polarons in quantum dots (QDs),²⁶" Which I believe should be ref. 37.

Response: Thank you for noticing and pointing this out to us. We have corrected the mistakes in the revised version of the manuscript.

Action: We have corrected the indicated reference numbers in the revised manuscript. In addition, we have thoroughly cross-checked the reference numbers throughout the manuscript.

- in regards to ref. 37, the authors should cite the work of Krawczyk et al "Non-thermal Lattice Disorder Observed in Photoexcited PbS Quantum Dots" J. Phys. Chem. C 2021, 125, 40 (2021). This latter reference is important to include as it points out the problem with ref. 37 (Lindeberg et al) in that the effect they observe is likely multiphoton ionization that has a much higher cross section at 400 nm than the band gap excitation. This point is important as they neglect multiphoton absorption and have to propose as surface hole state fully resonant with the band states. There is no way this would lead to a long lived trapped hole". I make this point as the authors need to think about their

excitation levels.

Response: Thank you for this valuable advice. We have accordingly cited the suggested study (ref. 42) as an example of light-induced lattice distortion of QDs in the revised manuscript.

Action: We have added the suggested reference as ref. 42 in the revised manuscript.

- Figure 3a. I could not understand the point of the photo. The authors should show where the field is applied and affect on transmission better. At least I could not understand to what effect other than a uniform filter the effect gave that would not be unusual. Please improve this figure to make the application more apparent.

Response: Thank you for your comment and we apologize for the lack of clarity. Figure 3a (Figure 4a in the revised manuscript) shows an illustration and a photograph of the IR-sensing devices. We wished to show that our discovery can be applied in invisible IR sensors; however, the employment of a transparent comb electrode seems to have made the figure confusing. Therefore, we have added the following Figures S11 (Figure R1) in the supporting information of the revised manuscript.

Action: We added the Figure R1 explaining the transparent devices in detail in the Figure S11 of supporting information of the revised version of the manuscript.

Figure R1. Photograph and schematic illustration of the transparent plasmonic IR sensor. CuS NC film was deposited on the surface of ITO comb electrode fabricated on the surface of the glass. Because both the CuS NC and ITO are transparent, the fabricated IR sensing device is transparent as well.

- Please remove reference Faraday's in the abstract namely "thereby expanding on Faraday's discovery". It is unnecessary and gives the impression that there was some unknown physics that is now solved by this work. The historical reference in the introduction is fine and makes the intended point.

Response: Thank you for the advice. We have deleted the indicated sentence from the abstract in the revised version of the manuscript.

Action: We have deleted the indicated sentence from the abstract in the revised version of the manuscript.

Major concern. I only really have one and that is that all of the noted effects could be due to changes in T with the very high peak powers used. The characterization and use of optical absorptivities all assumes weak perturbation in the 1-photon limit. However, the short excitation pulses used puts the excitation levels at 50 GW/cm^2 assuming 100 fs pulses. (Please give more details on the excitation pulses and whether 5 mJ/cm^2 was kept constant for 400 nm and 800 nm with error bars). There is no doubt in this reviewer's mind that there is multiphoton absorption occurring at these peak powers. The estimated T jump is far too low an estimate. I am not even sure how a local surface plasmon could have such a low absorptivity even in the weak perturbation limit.

The authors must measure the absorbed excitation at the different fluences used, and do a power dependence. This point is made specifically in light of the observation that above 8 mJ/cm^2 the excitation is no longer reversible. If the ΔT changes are as small as the authors estimate, then this onset of damage at 8 mJ/cm^2 makes no sense. Surely if they heat the films up by 11 degrees to 30 degrees above RT, they would not see the films change such that if their ΔT estimates were correct the process should be completely reversible to much higher excitation, especially give this ΔT is for the absorbed laser energy prior to diffusion. The actual ΔT under steady state KHz excitation in data collection would be much less and should be completely reversible.

Response: Thank you very much for your very important advice. We employed 400- and 800-nm laser with pulse width of 100fs for excitation. The laser with the fluence of 5 mJ/cm^2 was stable and almost constant (error bars, 800 nm: ~1%, 400 nm: ~2%).

According to your suggestion, we conducted laser fluence-dependency experiments for the electron diffraction intensity from the (110) planes (Figure R2). If we consider that the changes in the electron diffraction intensities are caused by simple laser-heating, the changes should increase linearly with laser fluence. However, the fluence dependency exhibited a linear change with a threshold, indicating that the changes in the electron diffraction intensities were caused not by simple laser-heating, but by non-thermal structural change (atomic rearrangements). We also confirmed that multiphoton absorption does not occur in this fluence range. A threshold was also observed for the power dependence of the IR light (800 nm)-responsive photoconductivity of the CuS NC film using the filtered Xe lamp as a light source (Figure R3). In photo-induced phase transition, the fluence threshold results from the energy barrier owing to latent heat (For example, *Phys. Rev. Lett.* 120, 207601 (2018); *Science* 346, 445 (2014); and *Phys. Rev. B* 83, 195120 (2011)). Such a threshold was observed as a feature of light-induced phase transition, but not of laser heating (*Phys. Rev. Lett.* 120, 207601 (2018)). The present observation of the threshold for the fluence-dependency of the atomic displacement allows us to conclude that the coherent motions are triggered by a cooperative lattice-orbital response.

Although we did not observe any significant damage in the current experiment ($<5 \text{ mJ/cm}^2$), the damage induced by a mechanism other than laser heating might be

observed under the higher fluence ($>8 \text{ mJ/m}^2$). The temperature rise was estimated from laser fluence, absorptivity, density, and specific heat, which are independent of the results of time-resolved electron diffraction measurements. The temperature rises at a fluence of 8 mJ/cm^2 was calculated to be $\sim 18 \text{ K}$, which is far from the melting point of CuS. As you addressed, the pulse duration of the laser is extremely short and multiphoton absorption or laser ablation might be dominant above the incident fluence of 8 mJ/cm^2 .

Action: We added an experiment to investigate laser fluence dependency in Figure S8 of the revised version of the supporting information document. In addition, we also added a description about the origin of light-stimulated ionic displacement in the 2nd paragraph of page 11 of the revised manuscript. We have also shown that the LSPR-induced photoconductivity shift also exhibits a threshold for light power dependence (Figure S12).

Figure R2. (a) Time evolution of the electron diffraction intensity from the (110) planes under fs-laser-excitation with different fluences at a wavelength of 800 nm. (b) Laser fluence-dependent shift in the electron diffraction intensity from the (110) planes.

Figure R3. Power dependence of IR light (800 nm)-responsive photoconductivity of the CuS NC film at a voltage of +1 V.

I will reserve judgement until this control is done but I suspect that the note effect is thermal in origin. At these excitation peak powers and strong electron-electron coupling in the 800 nm with respect to plasmon formation that the system is close to the formation of lattice breakdown or preplasma from many sources of multiphoton emitted electron. The absorption skin depth would then be more comparable to preplasma formation in nominally transparent materials at higher excitation. The change in lattice T could be much higher and the effect of resonant multiphoton ionization would also affect the lattice stability. The dual effect of fully resonant multiphoton ionization, collective coupling to the surface plasmon, could lead to very large T jumps sufficient to drive lattice displacements for the now highly perturbed lattice.

Response: Thank you very much for the insight. With the additional experiment, we concluded that the major process is light-stimulated Jahn-Teller effect because of the following three reasons.

1. As shown in Figure S8 (Figure R2) in the Supplementary Information, laser heating with 800-nm light caused a temperature increase of 11 K. The modulation in diffraction intensity induced by this is less than 1%. However, the decrease in intensity of the 1 1 0 diffraction ring upon the excitation of the LSPR band was ~6%. This fact indicates that the photo-induced intensity changes in the diffraction pattern of the CuS NCs could not be attributed to the simple photothermal Debye-Waller effect.

2. The fluence dependence of the intensity shift of the (011) plane exhibited a threshold. In photo-induced phase transition, the fluence threshold results from the energy barrier owing to the latent heat. Such a threshold was observed for light-induced phase transition, but not for laser heating (*Phys. Rev. Lett.* 120, 207601).

3. Figure 3c shows the LSPR-stimulated change in intensity of (110) planes: the intensity first decreases sharply, increases a little, and then becomes constant. However, thermal behavior often changes exponentially at a certain point, with the time constant to return to the original state being heat dissipation, which can range from nano to microseconds (*J. Phys. Chem. B*, 2006, 110, 50, 25308–25313). Therefore, the unique change in intensity of (110) planes stimulated by LSPR suggests that the observed behavior is certainly not thermal, indicating the occurrence of non-thermal structural change.

However, the thermal effect cannot be ignored in light-stimulated phase transition, as latent heat is required to be supplied for phase transition. The plasmonic heating of CuS NCs may therefore harness the phase transition in our systems. We plan to conduct a detailed experiment in the future to investigate and clarify the role of plasmonic heating.

I want to assure the authors I strongly support publication of this work in Nature Comm. I think I would not be doing my job if I did not ask for a power dependence on the time resolve diffraction data as well as transient absorption spectra. The absorption of the excitation pulse must be measured directly and not use calculated values based on weak perturbation optical properties.

This manuscript represents a beautiful set of experiments with clearly a very large lattice change with ionic motions strongly modulating the CuS nanoparticle properties. I fully support publication and hope the authors find the above suggestions helpful.

Response: We thank the reviewer once again for their valuable insight and suggestions, which we found extremely helpful. From the results of the additional experiment and after careful consideration, we believe that the LSPR-induced ionic displacement is caused by the Jahn–Teller effect rather than the thermal effect. We hope that the additional experiment made the discussion clear. We also thank you for your positive evaluation of the importance of our work.

Reviewer #2 (Remarks to the Author):

The present manuscript discusses an interesting and timely topic, a plasmon-resonance induced Jahn-Teller distortion. The authors report on an interesting series of experiments performing state of the art investigations. Upon exciting a film of CuS nanoparticles, a different response is reported depending upon the laser wavelength. A pronounced change of optical properties is observed for an excitation below (!) the band gap. This is attributed to a change in atomic arrangement. Subsequently, the authors characterize this change in atomic arrangement and the time scale of the return to the initial state. Finally, the authors demonstrate a potential application by showing a photoconductivity change.

This is an interesting sequence of experimental data and I hence recommend acceptance of this manuscript after making several improvements.

Response: Thank you for your valuable comments, suggestions, and insight as well as positive feedback. We have addressed all comments. We hope that the explanation and revision of our work are clear and sufficient.

These comments and suggestions are listed in the following chronologically:

a) I found the first two paragraphs of the main text quite vague. For example, what are the limits of plasmonics?

Response: Thank you for your feedback and we apologize for the lack of clarity. The light-to-matter interaction in LSPR of conventional noble metal nanocrystals has been limited to the stimulation of the collective mode. No exceptional relaxation processes have been found in the history of plasmonics. Of course, collective mode stimulation is known to be the origin of various important applications of plasmonic materials, while atom/ion displacement can have a great impact on the material over time. Therefore, the development of LSPR-induced stimulation of metastable atom/ion displacement will further expand the possibility of LSPR.

Action: We have rewritten the indicated paragraph to present the discussion more clearly in the revised version of manuscript.

How will they be pushed by the present investigation?

Response: Thank you for your question. Thus far, it has been considered that LSPR cannot induce metastable atom/ion displacement, which causes dramatic changes in material properties. Therefore, the discovery that LSPR can indeed induce the stimulation of metastable atom/ion displacement will expand the possibilities for application of LSPR.

Action: We have rewritten the indicated paragraph to present the discussion more clearly in the revised version of the manuscript.

Which elements of what relaxation mechanism are unclear?

Response: Thank you for your question. CuS NCs exhibit an inherently slow LSPR relaxation in the microsecond region (lifetime = 1.7 μ s). This unique delay in relaxation is indicated by the laser flash photolysis results of CuS NCs, which indicate an extraordinarily slow LSPR-bleach recovery. However, the origin has been unclear.

Action: We have rewritten the indicated paragraph to present the discussion more clearly in the last paragraph of page 2 in the revised version of the manuscript.

b) On p.5, the authors should say that their NCs are platelets, then the numbers for the lateral diameter and thickness are easier to understand.

Response: Thank you for the suggestion. We have accordingly described the CuS NCs as having a plate-like structure.

Action: We revised the indicated sentence (sentence 1, paragraph 1, page 5 in the revised manuscript) as follows: “CuS NCs with plate-like structures were synthesised according to a previous publication.”

c) The authors might want to replace the phrase ‘global phenomenon’ by ‘collective atomic rearrangement’

Response: Thank you for the suggestion. We have made the change in the revised manuscript.

Action: We have replaced the phrase “global phenomenon” with “collective atomic rearrangement” in the revised version of the manuscript.

d) The caption of figure 2 does not say anything about the laser pulse needed (pulse length and power).

Response: Thank you for your indication and we apologize for the oversight. We used an Xe lamp (Cermax PE300BF) for the SAED under light irradiation. Xe lamp irradiates over a broad spectrum, while the laser experiment clearly shows that only IR irradiation causes ionic displacement.

Action: We have revised the first sentence of the figure caption as follows, “Electron diffraction pattern of CuS under light irradiation from Xe lamp with an intensity of ca. 140 mW/cm² (Cermax PE300BF).”

e) It is confusing that the label: figure 2, is shown twice. Place label all figures sequentially from 1 to 4.

Response: We apologise for the oversight. Thank you for pointing it out to us.

Action: We have corrected the mistakes in figure numbering in the revised version of the manuscript.

f) The authors argue on p. 11 that the decrease of diffraction intensity was not shown for bandgap excitation, but their data in figure 2.b show a similar change in atomic rearrangement.

These explain this apparent contradiction.

Response: We are sorry for our mistakes in labeling Figures 3b and c and causing confusion. Figure 3b indicates the shift in the Q value, while Figure 3c shows the change in the diffraction intensity caused by laser irradiation.

Action: We have added “Figure 3c” in the revised version of the manuscript to avoid confusing the readers.

g) In figure 3.b (should be 4.b), the caption does not specific the light intensity needed to show the effect. It also would be nice to know if the effect depends upon the light intensity in a linear fashion.

Response: Thank you very much for your very important advice. According to your suggestion, we conducted laser fluence-dependency experiments for the electron diffraction intensity from the (110) planes (Figure R4). If we consider that the changes in the electron diffraction intensities are caused by simple laser-heating, the changes should increase linearly with laser fluence. However, the fluence dependency exhibited a linear change with a threshold, indicating that the changes in the electron diffraction intensities were caused not by simple laser-heating, but by non-thermal structural change (atomic rearrangements). We also confirmed that multiphoton absorption does not occur in this fluence range. In photo-induced phase transition, the fluence threshold results from the energy barrier owing to latent heat (For example, *Phys. Rev. Lett.* 120, 207601 (2018); *Science* 346, 445 (2014); and *Phys. Rev. B* 83, 195120 (2011)). Such a threshold was observed as a feature of light-induced phase transition, but not of laser heating (*Phys. Rev. Lett.* 120, 207601 (2018)). The present observation of the threshold for the fluence-dependency of the atomic displacement allows us to conclude that the coherent motions are triggered by a cooperative lattice–orbital response.

The threshold was also observed for the IR-induced shift of conductivity of CuS film (Figure R5).

Action: We added an experiment to investigate laser fluence dependency in Figure S8 of the revised version of the supporting information document. In addition, we also added a description about the origin of light-stimulated ionic displacement in the 2nd paragraph of page 11 of the revised manuscript. We have also shown that the LSPR-induced photoconductivity shift also exhibits a threshold for light power dependence (Figure R5) in the Figure S12 of the revised version of the supporting information document.

For Figure 4c in the previous version of manuscript, we did not take the laser power dependence into the account. Therefore, we replace the Y axis of Figure 4c by the current shift divided by illuminated light power in the revised version of manuscript. Since the new Y axis also reproduced the LSPR of CuS NCs, we believe that the change does not affect our discussion.

Figure R4. (a) Time evolution of the electron diffraction intensity from the (110) planes under fs-laser-excitation with different fluences at a wavelength of 800 nm. (b) Laser fluence-dependent shift in the electron diffraction intensity from the (110) planes.

Figure R5. Power dependence of IR light (800 nm)-responsive photoconductivity of the CuS NC film at a voltage of +1 V.

h) The authors argue that the crystal structure shift is extremely long. Yet, this shift only lasts for 50 ps. Why is this extremely long? Please explain this in more detail.

Response: We apologize for our presentation being confusing. The lifetime of 50 ps suggests the dynamics of ionic displacement just after excitation. As shown in Figure R6, the decrease in intensity was not completely recovered at 500 ps after laser excitation, indicating that the relaxation of ionic displacement requires a much longer time. In our previous work (*Nat Commun* 9, 2314 (2018)), we discovered that the relaxation of LSPR of CuS involves anomalously long-lived components ($\sim 1.7 \mu\text{s}$)

(Figure R7). We therefore considered that the slow recovery of atomic displacement corresponds to the slow relaxation of LSPR (Figure R8).

Figure R6. Time evolution of the electron diffraction intensity from the (110) planes under fs-laser excitation at wavelengths of 400 and 800 nm.

Figure R7. Time evolution of the recovery of LSPR of CuS observed in the ns-transient absorption measurement (*Nat Commun* 9, 2314 (2018)).

Figure R8. LSPR relaxation process of CuS proceeds through Landau damping, carrier relaxation, and an unknown delayed relaxation corresponding to the time domains of < 270 fs, 110 ps, and 1.7 μ s, respectively (*Nat Commun* 9, 2314 (2018)). The ionic displacements associated with LSPR excitation cause the delayed LSPR relaxation shown in the subsequent figures.

authors have responded in sufficient detail to the questions, comments and criticism of the two reviewers. This is clearly appreciated. Hence, I suggest to accept the manuscript.

Response: Thank you for your valuable input and positive feedback. We hope that the explanation and revision of our work are sufficient and clear.

At the same time, I would like to summarize where my personal opinion and belief is not compatible with the models and explanations suggested in the present manuscript. I leave it up to the authors completely, if they respond to these comments or prefer to ignore them.

Response: We have replied to each comment individually, and value the feedback immensely.

a) Tetrahedral Ge:

Indeed, many groups have adapted initially adopted the tetrahedral Ge model by Tominaga, Kolobov and coworkers to explain the atomic arrangement in amorphous phase change materials like GeTe or Ge₂Sb₂Te₅. There is no doubt that this model can account for a number of experimental observations, such as the pronounced property contrast between the amorphous and the crystalline state. Yet, subsequent studies have shown both experimentally and theoretically that another model, where the amorphous material is characterized also by an octahedral atomic arrangement (like the crystal) provides a better explanation of the properties of the amorphous system. Aging is one of these phenomena. It appears as if amorphous GeTe ‘ages’ and upon doing so develops properties which are even more dissimilar from the crystalline phase. This is explained easily if one assumes an increasing Peierls distortion in the amorphous phase, in line with the observed changes of the optical properties. Simulations show rather unequivocally that in the amorphous phase upon relaxation

the number of tetrahedral sites decreases and the distortion of the octahedral sites increases.

The authors seem to indicate that tetrahedral Ge is stabilized in Te rich compounds such as GeTe₂. This is indeed a possible scenario. Yet, under Te-rich conditions it seems questionable if an octahedral arrangement would be a plausible scenario as the lowest energy state.

Finally, the Stanford group (E. Pop and coworkers) that recently also reported on superior switching in Sb₂Te₃ / GST₂₂₅ superlattices and commented that this improvement is only achieved for very thin films of Sb₂Te₃, implying that it is the finite thickness of the Sb₂Te₃ layer which is crucial for the superior behavior of these superlattices [*Nano Lett.* 2022, 22, 15, 6285–6291]. This observation cannot easily be explained by the tetrahedral Ge model discussed here.

b) Sb₂Te₃ (2D)

The authors seem to argue that Sb₂Te₃ is a 2D material. Sb₂Te₃ is clearly more anisotropic than GeTe and has a layered structure. Yet, the interaction across the Te – Te gaps is much larger than for typical layered materials like graphite or hexagonal BN. Hence, the label as a 2D material seems misleading. In this context the authors also compare chemisorption vs. physisorption. The concept of chemisorption would not be compatible with true 2D behavior.

Also, recent publications have shown that the layer spacing in Sb₂Te₃ is too close compared with true van der Waals bonded systems, which show a larger distance across the gap.

Finally, I just note in passing that the phrase *inherent vacancy* might be replaced by the word '*stoichiometric vacancy*'. In GeSb₂Te₄, this is one stoichiometric vacancy within each unit cell, nevertheless the material has a band gap and is a semiconductor. *Excess vacancies* would make this material metallic, if sufficiently high numbers of excess vacancies exist. The authors might want to consider these comments, but I do not request any further change of their manuscript.

Response: Thank you very much for your interesting insight, which we have carefully considered. However, it is difficult to add a description of the suggested compound in the revised version of the manuscript. Because the suggested compounds are not plasmonic and the crystal structure is different, we are afraid that the additional description would divert our discussion about LSPR-induced ionic displacement.

We greatly appreciate the important suggestion, however, and would like to investigate the light-stimulated phase shift of the suggested material in the future.

Reviewer #3 (Remarks to the Author):

In this manuscript, it is confirmed that localised surface plasmon resonance (LSPR) induced the cooperative Jahn Teller effect in covellite CuS nanocrystals (NCs), causing metastable displacive ion movements with several structural analysis and pump-probe dynamics and supported with suitable calculations.

Based on the existing studies, this research has been well conducted in a groundbreaking way. Moreover, there are attempts to overcome the limitations of the existing studies such as failure to track the ionic coordination or elucidate the mechanism caused by LSPR.

However, additional explanations and experiments to reinforce the arguments of the manuscript are still needed.

Response: Thank you for your valuable comments, suggestions, and insight as well as positive feedback. We have addressed all comments. We hope that the explanation and revision of our work are clear and sufficient.

See attached file

In this manuscript, it is confirmed that localised surface plasmon resonance (LSPR) induced the cooperative Jahn Teller effect in covellite CuS nanocrystals (NCs), causing metastable displacive ion movements with structural analysis and pump-probe dynamics and supported with suitable calculation. However, some revisions on manuscript are needed. Questions about the manuscript are as follows:

Comments.

1. The reason why the plasmon excitation and relaxation processes of CuS, which is not a noble metal, are all on the ps scale, but the combined time scale is us scale is needed.

Response: Thank you for your comment. The results of ultrafast time-resolved electron diffraction analyses and theoretical calculations of semiconductive plasmonic CuS NCs indicated that the metastable displacive ion movements caused by the LSPR-induced cooperative Jahn–Teller effect caused a delay in the relaxation of LSPR in the microsecond region (Figure R9).

Figure R9. LSPR relaxation process of CuS proceeds through Landau damping, carrier

relaxation, and an unknown delayed relaxation corresponding to the time domains of < 270 fs, 110 ps, and 1.7 μ s, respectively (*Nat. Commun.* 9, 2314 (2018)). The ionic displacements associated with LSPR excitation cause the delayed LSPR relaxation shown in the subsequent figures.

2. In the process of claiming that the ionic displacement observed in Figure 3.d is a cooperative phenomenon, the diffraction width was arbitrarily designated and it was stated that the width did not change. Additional evidence is needed to support the above claim.

Response: Thank you for your insight. We have investigated the FWHM of the diffraction spot of SAED (Figure R10). The FWHM of the diffraction spot of SAED was not observed by the light irradiation. This indicates that the long-range structural periodicity of the system was maintained during the light-stimulated ionic displacement.

In addition, we conducted experiments to investigate the laser fluence-dependency of the electron diffraction intensity of the (110) planes (Figure R11). If we consider that the changes in electron diffraction intensity are caused by simple laser-heating, the changes should increase linearly with laser fluence. However, the fluence dependency exhibited a linear change with a threshold, which means that the changes in the electron diffraction intensities are caused not by simple laser-heating, but by non-thermal structural change (atomic rearrangements). We also confirmed that multiphoton absorption does not occur in this fluence range. A threshold was also observed for the power dependence of the IR light (800 nm)-responsive photoconductivity of the CuS NC film. In photo-induced phase transition, the fluence threshold results from the energy barrier owing to the latent heat (For example, *Phys. Rev. Lett.* 120, 207601(2018); *Science* 346, 445 (2014); *Phys. Rev. B* 83, 195120(2011)). The present observation of the threshold for the fluence-dependency of atomic displacement allows us to conclude that the coherent motions are triggered by a cooperative lattice-orbital response. The threshold for the irradiated light intensity was also observed for the IR-induced shift of conductivity of the CuS film (Figure R12).

Action: We added the FWHM of diffraction spots of SAED with and without light illumination as Figure S5 in the revised version of the Supporting Information document. Furthermore, we added the description about the shift of FWHM by light irradiation in the last paragraph of page 7 in the revised version of manuscript.

We added an experiment to investigate the laser fluence dependency, the results of which are shown in Figure S7 of the revised version of the Supporting Information document. In addition, we also added a description of the origin of light-stimulated ionic displacement in the second paragraph of page 11 of the revised version of the manuscript.

We have also shown that the LSPR-induced photoconductivity shift also exhibits a threshold for light power dependency in Figure S12 of the revised version of the Supplementary Information document.

Figure R10. FWHM of diffraction spots of SAED with and without light illumination.

Figure R11. (a) Time evolution of the electron diffraction intensity from the (110) planes under fs-laser-excitation with different fluences at a wavelength of 800 nm. (b) Laser fluence-dependent shift in the electron diffraction intensity from the (110) planes.

Figure R12. Power dependence of IR light (800 nm)-responsive photoconductivity of the CuS NC film at a voltage of +1 V.

- It is stated that LSPR excitation in CuS did not change the long-range structural periodicity of the system. Is there any data such as SAED pattern or X-ray diffraction which can support the above invariability?

Response: Thank you very much for your very important question. In order to answer it, we investigated the FWHM of diffraction spots of CuS NC with and without light irradiation (Figure R10). The FWHM did not change because of light irradiation, indicating that the long-range structural periodicity of the system was maintained.

4. In this study, laser excitation and photoconductivity measurements were performed with 400nm (which is correspond to band transition region) and the 800nm wavelength (which is correspond to LSPR region) to show that the cooperative Jahn-Teller effect is due to LSPR. Are there other data sets where the same experimental splits were performed with the wavelength value of the 600nm (middle regime) or 1000nm (high value in LSPR regime)?

Response: Thank you very much for your important advice. We have investigated the time evolution of the peak-position and intensity of the diffraction ring after excitation with 1053- and 527-nm laser light irradiation (Figure R13). For this experiment, we used a Q-switched nanosecond laser with an irradiation duration of 10 ns. It was clear that laser excitation (via 1053- and 527-nm light) caused a negative shift of the Q -value of the (110) plane (Figures R13 a and c), indicating that the CuS NCs underwent lattice expansion because of the laser heating. Furthermore, the diffraction intensity decreased, which was observed under LSPR excitation (1053-nm light) but not under bandgap excitation (527-nm light), indicating that LSPR excitation induced ionic displacements in the unit cell. The obtained result agreed well with the results of 400-nm and 800-nm excitation.

Action: We added the time evolution of the peak position and intensity of the diffraction ring after excitation with 1053- and 527-nm laser light irradiation in Figure S13 in the revised version of the Supporting Information document. Furthermore, we added the description of the excitation laser wavelength dependence in the 1st paragraph of page 11 in the revised version of the manuscript.

Figure R13. **a** Time evolution of the Q -value from the (110) planes under fs-laser excitation at a wavelength of 1053 nm. **b** Time evolution of the electron diffraction intensity from the (110) planes under fs-laser excitation at a wavelength of 1053 nm. **c** Time evolution of the Q -value from the (110) planes under fs-laser excitation at a wavelength of 523 nm. **d** Time evolution of the electron diffraction intensity from the (110) planes under fs-laser excitation at a wavelength of 523 nm.

REVIEWER COMMENTS

Reviewer #1 (Remarks to the Author):

The authors have dutifully responded to my concerns and have done a control experiment where they have studied the fluence dependence. They argue that the observed fluence dependence is not consistent with a thermally driven lattice/phase transition (that is not well characterized by pressure or T studies). They provide the following arguments to which I reply below each point.

"1. As shown in Figure S8 (Figure R2) in the Supplementary Information, laser heating with 800-nm light caused a temperature increase of 11 K. The modulation in diffraction intensity induced by this is less than 1%. However, the decrease in intensity of the 1 1 0 diffraction ring upon the excitation of the LSPR band was ~6%. This fact indicates that the photo-induced intensity changes in the diffraction pattern of the CuS NCs could not be attributed to the simple photothermal Debye–Waller effect."

This argument is incorrect. I have gone over the T calculation and agree with it – if and only if the laser excitation is uniformly exciting the CuS nanodot/film of 60 nm. The very fact that CuS has a localized surface plasmon means that the field interaction is strongly confined to the surface region. Some estimate of the skin thickness of the localized surface plasmon resonance (LSPR) is needed. The temperature and ensuing nonlinear thermal response would come from this layer. This T change could be enormous. More importantly, the enhancement is so strong in this region, there is every reason to believe that multiphoton processes are occurring using femtosecond laser pulse excitation at 50 GW/cm² for such strong resonances in the surface region. There will certainly be photoemission (the authors should check this).

The additional problem is their other control studies in response to reviewer 3 where they do not see the same effect at 1.053 micron excitation using nanosecond pulses even though this wavelength is clearly in the heart of the LSPR. The difference is that these 1.053 micron excitation studies used nanosecond pulses not femtosecond pulses. (I note here that the figure caption R13 incorrectly states femtosecond pulses whereas the main text and rebuttal state 10 nanosecond excitation.) The difference in peak power is 100,000x. At low peak power, even with similar fluence, the effect vanishes even for more strongly resonant conditions with the LSPR at 1.053 micron excitation than 800 nm. This observation points to a peak power effect.

It could be argued that the LSPR excitation damps faster than 10 ns (Landau damping) so that the effect can only be observed with fs pulses. The authors really must do a pulse width dependence to see if the effect scales as expected...linearly with a threshold for longer pulses still within the damping time scale. It will be very important to see if there is photoemission from the surface as a function of excitation duration and peak power, which can be done by simply shadow imaging the surface with the electron probe. This control involves rotating the crystal but this should be simple enough if the fs ediff instrument has a 90 degree viewing port or enough space to enable simple off axis excitation.

Given the enormous surface fields present at the CuS surface, by virtue of the fact it exhibits a localized surface plasmon, it is really important to rule out peak power and multiphoton ionization effects that destabilize the lattice potential to give the observed effect.

"2. The fluence dependence of the intensity shift of the (011) plane exhibited a threshold.

In photo-induced phase transition, the fluence threshold results from the energy barrier owing to the latent heat. Such a threshold was observed for light-induced phase transition, but not for laser heating (Phys. Rev. Lett. 120, 207601)."

I respectfully disagree. Without going into details, I think the authors are aware of work on nonthermal phase transitions studied with fs electron diffraction to directly observe the electronically driven atomic motions. The effect always involves exactly the same energy deposited as would be needed for a thermally driven transition. The same amount of work in lattice restructuring is needed. The difference is the time scale of the motions, assigned to electronically driven motions, always occurs faster than thermalization processes. This separation of time scales does not seem to be operating here. Also, there is always a threshold for both thermal and electronically driven phase transitions for the same reason given, i.e., the process must overcome the latent heat or work both in atomic displacement and overcoming the barrier to nucleation, the transition barrier between different phases. The observation of a threshold does not rule out thermally or my major concern of multiphoton electron emission in the surface region.

"3. Figure 3c shows the LSPR-stimulated change in intensity of (110) planes: the intensity first decreases sharply, increases a little, and then becomes constant. However, thermal behavior often changes exponentially at a certain point, with the time constant to return to the original state being heat dissipation, which can range from nano to microseconds (J. Phys. Chem. B, 2006, 110, 50, 25308–25313). Therefore, the unique change in intensity of (110) planes stimulated by LSPR suggests that the observed

behavior is certainly not thermal, indicating the occurrence of non-thermal structural change."

I agree to a certain extent here but the nonthermal effect of concern is peak power dependent multiphoton emission rather than LSPR effects leading to ion displacement. (Also I note that the decay time is similar to thermal dissipation for in vacuum conditions.)

"However, the thermal effect cannot be ignored in light-stimulated phase transition, as latent heat is required to be supplied for phase transition. The plasmonic heating of CuS NCs may therefore harness the phase transition in our systems. We plan to conduct a detailed experiment in the future to investigate and clarify the role of plasmonic heating."

I highly recommend the authors do the peak power dependence by varying the pulse duration under constant fluence (within the limits of Landau damping) and most important look for surface photoemission. I am sure this effect is operating. It is only a question about how big an effect this has on the lattice potential. I can well imagine this being a much bigger effect than any thermal process (even with extremely high surface temperatures in the surface plasmon region...as this too will lead to thermoionic emission).

Please note, I strongly support publication of this work. I hope the authors will do the above controls. I apologize if I was not clear enough in my original report to bring this issue to their full attention. I think a compromise could be made in which the authors estimate the magnitude of the surface fields in the presence of the LSPR and degree of photoemission. It would be instructive for them to look at fs studies of photoemission due to surface plasmons of Cu, some of the early published work on this topic.

Reviewer #2 (Remarks to the Author):

The authors have carefully considered my comments and suggestions. I hence suggest acceptance of the manuscript in its present form.

Reviewer #3 (Remarks to the Author):

Through the revision process, the authors clearly answer with sufficient explanations as follows and data sets for suggested questions:

The time scale of the plasmon excitation and relaxation process are explained through schematics and proper reference.

It is properly answered through the SAED pattern that the ionic displacement corresponds to the cooperative phenomenon and that the LSPR excitation of CuS does not change in the long-range period.

Regarding the photoconductivity experiment, it is answered faithfully by presenting data sets where wavelength corresponding to the inside and outside of the two values previously presented. (400 and 800 nm)

Point-by-point responses to the reviewers' comments for NCOMMS-22-

REVIEWER COMMENTS

Reviewer #1 (Remarks to the Author):

The authors have dutifully responded to my concerns and have done a control experiment where they have studied the fluence dependence. They argue that the observed fluence dependence is not consistent with a thermally driven lattice/phase transition (that is not well characterized by pressure or T studies). They provide the following arguments to which I reply below each point.

Answer: We have dutifully extended the additional works: Time-resolved electron diffraction measurements with longer duration for excitation pulse (without changing the fluence), shadow imaging experiments with shorter and longer duration of excitation pulses, and X-ray diffraction measurements by changing the temperature at the synchrotron facility. The results confirmed that the observed phenomenon is induced by atomic displacements from photoinduced LSPR effects, and the contribution of the electron scattering by the surface plasmon is negligibly small.

We thank you for your comments and believe that these additional experiments have enriched the manuscript. We are also grateful for your strong support of our study. The replies to each response are appended below.

"1. As shown in Figure S8 (Figure R2) in the Supplementary Information, laser heating with 800-nm light caused a temperature increase of 11 K. The modulation in diffraction intensity induced by this is less than 1%. However, the decrease in intensity of the 1 1 0 diffraction ring upon the excitation of the LSPR band was ~6%. This fact indicates that th

photo-induced intensity changes in the diffraction pattern of the CuS NCs could not be attributed to the simple photothermal Debye–Waller effect."

This argument is incorrect. I have gone over the T calculation and agree with it – if and only if the laser excitation is uniformly exciting the CuS nanodot/film of 60 nm. The very fact that CuS has a localized surface plasmon means that the field interaction is strongly confined to the surface region. Some estimate of the skin thickness of the localized surface plasmon resonance (LSPR) is needed. The temperature and ensuing nonlinear thermal response would come from this layer. This T change could be enormous. More importantly, the enhancement is so strong in this region, there is every reason to believe that multiphoton processes are occurring using femtosecond laser pulse excitation at 50 GW/cm² for such strong resonances in the surface region. There will certainly be photoemission (the authors should check this).

Answer: We are grateful for reminding us of the importance of the surface effects. Regarding the former portion of this comment and according to the comments, we have extended the time-resolved electron diffraction measurements by changing the excitation pulse duration (110 fs and 260 fs) to introduce 200-mm-long N-BK7 media in the optical pump line, as shown in Figure R1 (S10 of the Supplementary Information). The intensity changes from the (110) plane did not change with pulse duration, indicating that the effect of multiphoton absorption on the surface area is negligible. If multiphoton ionization had occurred, the significant change should be observed depending on the pulse width because the pulse width was changed by 2.5 times, but there was no change (Δ intensity remains -4% at both 110fs and 260fs). Therefore, we believe that the possibility of multiphoton ionization can be ignored.

This result is logical since the structural changes occur even under CW light, as shown in Figure 2a in the main manuscript. We used the Xe lamp and long-pass filter ($\lambda > 800$ nm) for the light-stimulated intensity changes with respect to the electron diffraction of the CuS nanocrystal (NC) based on TEM. Generally, light sources with weak fluences, such as the Xe lamp, could not cause the non-linear two-photon excitation. This fact also supports our hypothesis.

Figure R1. (a) Time evolution of the electron diffraction intensity from the (110) planes under fs-laser excitation (wavelength: 800 nm, fluence: 5 mJ cm⁻²) under different pulse durations. For the time-resolved electron diffraction experiments at a pulse duration of 260 fs, we introduced N-BK7 media (length of 200 mm) in the pump arm to extend the pulse duration by dispersion. (b) Intensity shifts as a function of pump pulse duration derived from (a). The intensity shifts at the pulse durations of 110 and 260 fs for the pump pulse are identical, which suggests that multiphoton absorption does not occur in this fluence range. This result is also consistent with the fact that the intensity shifts in the electron diffraction intensity are changed linearly by the laser fluence (Figure S9).

Action: We added Figure R1 (as a Figure S10) in the revised version of the Supplementary Information. Additionally, we added related description on page 11, third paragraph of the revised version of the manuscript.

The additional problem is their other control studies in response to reviewer 3 where they do not see the same effect at 1.053 micron excitation using nanosecond pulses even though this wavelength is clearly in the heart of the LSPR. The difference is that these 1.053 micron excitation studies used nanosecond pulses not femtosecond pulses. (I note here that the figure caption R13 incorrectly states femtosecond pulses whereas the main text and rebuttal state 10 nanosecond excitation.) The difference in peak power is 100,000x. At low peak power, even with similar fluence, the effect vanishes even for more strongly resonant conditions with the LSPR at 1.053 micron excitation than 800 nm. This observation points to a peak power effect.

Answer: We apologise for the typographical errors and poor presentation that may have been confusing, which have been revised accordingly. The time-resolved electron diffraction measurements at an excitation wavelength of 1053 nm were performed using a nanosecond laser (duration of 10 ns). The 1053-nm excitation decreased the intensity of the (110) diffraction peak, while a 527-nm excitation did not (Figure R2). This result indicates that the excitation of LSPR triggers a cooperative ionic displacement. Since we used a different laser with a different wavelength, the noise level in these experiments was higher than that in the femto-to-picosecond time-resolved experiments, and the photoexcitation level could not be set to 5 mJ/cm² due to the divergence of the laser. However, a slight decrease was observed in the 1053-nm excitation, while no decrease was observed in the 527-nm excitation. The lattice expansion occurred for the excitation wavelengths of 527 and 1053 nm.

Figure R2. Illustration showing the relationship between excitation wavelength and shift in the intensity of the 1 1 0 diffraction peak.

Action: We have corrected the typographical error and revised the caption of Figure S8 in the revised version of the Supplementary Information.

It could be argued that the LSPR excitation damps faster than 10 ns (Landau damping) so that the effect can only be observed with fs pulses. The authors really must do a pulse width dependence to see if the effect scales as expected...linearly with a threshold for longer pulses still within the damping time scale. It will be very important to see if there is photoemission from the surface as a function of excitation duration and peak power, which can be done by simply shadow imaging the surface with the electron probe. This control involves rotating the crystal but this should be simple enough if the fs ediff instrument has a 90 degree viewing port or enough space to enable simple off axis excitation.

Given the enormous surface fields present at the CuS surface, by virtue of the fact it exhibits a localized surface plasmon, it is really important to rule out peak power and multiphoton ionization effects that destabilize the lattice potential to give the observed effect.

Answer: We performed shadow imaging experiments with shorter and longer optical pulses. As expected, the surface effects were observed after photoexcitation, as shown in Figure S10; however, the observed electron scattering was approximately 0.5%, which is significantly smaller than the intensity decreases observed in the time-resolved electron diffraction measurements (4–6%). The scattering effect did not change with the optical pulse duration, suggesting that the multiple photoexcitation and surface plasmon effects are significantly smaller than the photoinduced atomic displacements in CuS nanocrystals (NCs). A discussion on the thermal effects was included in response to the comments.

Figure R3. (a) Electron image used for the observation of photoemission from the sample surface. The CuS sample was set parallel to the electron beam. The sample cut half the electron beam (in diameter), and half of the electron beam travelled through and was detected by the CCD camera downstream. A semicircle pattern was observed on the CCD camera, and the straight line part of the semicircle surrounded by the yellow box shows the surface area. The wavelength and fluence of the pump optical pulse were fixed at 800 nm and 5 mJ cm^{-2} , respectively. (b) Time evolution of the electron intensity changes on the surface area. The electron intensity decreases slightly ($\sim 0.4\%$) after

$t = 0$ due to scattering via the LSPR effect. The scattering does not change with pulse duration, indicating that multiphoton absorption does not occur in this fluence range. The differential images at -20 ps (c) and +40 ps (d) at the pulse duration of 110 fs for the pump pulse and at -20 ps (e) and +40 ps (f) at the pulse duration of 260 fs. The slight decreases on the surface area and slight increases at the position away from the surface were observed after $t = 0$ for the pump pulse durations of 110 fs (d) and 260 fs (f).

Action: We added Figure R3 as Figure S11 in the revised version of the Supplementary Information.

"2. The fluence dependence of the intensity shift of the (011) plane exhibited a threshold. In photo-induced phase transition, the fluence threshold results from the energy barrier owing to the latent heat. Such a threshold was observed for light-induced phase transition, but not for laser heating (Phys. Rev. Lett. 120, 207601)."

I respectfully disagree. Without going into details, I think the authors are aware of work on nonthermal phase transitions studied with fs electron diffraction to directly observe the electronically driven atomic motions. The effect always involves exactly the same energy deposited as would be needed for a thermally driven transition. The same amount of work in lattice restructuring is needed. The difference is the time scale of the motions, assigned to electronically driven motions, always occurs faster than thermalization processes. This separation of time scales does not seem to be operating here. Also, there is always a threshold for both thermal and electronically driven phase transitions for the same reason given, i.e., the process must overcome the latent heat or work both in atomic displacement and overcoming the barrier to nucleation, the transition barrier between different phases. The observation of a threshold does not rule out thermally or my major concern of multiphoton electron emission in the surface region.

Answer: We are thankful for reminding us the work on nonthermal phase transition studies, and your comment is accurate. For the induction of electronically driven atomic motions, the same energy deposition is required as that required for a thermally driven transition. Therefore, for the observation of the threshold of fluence dependence, the thermal effects cannot be excluded.

Multiphoton electron emission from the surface region can be excluded based on the discussion above (the pulse duration dependence of time-resolved electron diffraction and shadow imaging experiments) (Figure R1). If the surface area is thermalised, while the inner area is unthermalised, the width of the diffraction peak will broaden because the diffraction peak from the surface area shifts without changing the diffraction peak from the inner area. However, peak broadening was not observed (Figure 3 in the main manuscript). Hence, we can assumed homogeneously photoexcited CuS NCs. Since the absorptivity of the CuS NC film is considerably weak, photoexcitation at 800 nm and 400 nm (5 mJ cm^{-2}) can increase the temperature by a few tens of Kelvin (10–30 K). The thermally driven intensity change in these temperature changes was less than 1%. The near-IR excitation (800 nm) induces LSPR transition, and near-UV excitation (400 nm) induces interband transition, respectively. The near-UV photoexcitation did not induce intensity change, while near-IR photoexcitation with the same fluence induced a 4–6% intensity decrease in the electron diffraction. This suggests that the observed intensity change in near-IR photoexcitation can be identical to the photoinduced phenomena via the LSPR transition.

The estimation that temperature changes of a few tens of Kelvin induce less than 1% intensity change in diffraction intensity was based on the Debye–Waller effects from the physical properties of bulk materials and not nanocrystalline materials. To estimate the Debye–Waller effects, we performed X-ray diffraction measurements by changing the temperature at the synchrotron facility (Aichi SR). Since the X-ray diffraction intensity changes of CuS NCs are significantly weak, they could not be measured using a laboratory-top X-ray diffractometer. Therefore, we measured the temperature dependence at the synchrotron facility. The diffraction patterns at various temperatures and the peak intensity changes are given in Figure R4. According to the figure, a few tens of Kelvin

induce ~1% change in the diffraction intensity.

Figure R4. Estimation of photo-thermal effects on CuS nanocrystals by measuring temperature dependence of X-ray diffraction patterns at the Aichi Synchrotron Radiation Center (Beam line: BL5S2). The X-ray wavelength of the synchrotron radiation was 0.7 Å. (a) X-ray diffraction patterns at various temperatures. (b) Enlarged view of the X-ray diffraction patterns at the (110) plane shown by black arrows. (c) Peak intensity shifts as a function of temperature.

Action: We added Figure R4 as Figure S7 to the revised version of the Supplementary Information.

"3. Figure 3c shows the LSPR-stimulated change in intensity of (110) planes: the intensity first decreases sharply, increases a little, and then becomes constant. However, thermal behavior often changes exponentially at a certain point, with the time constant to return to the original state being heat dissipation, which can range from nano to microseconds (J. Phys. Chem. B, 2006, 110, 50, 25308–25313). Therefore, the unique change in intensity of (110) planes stimulated by LSPR suggests that the observed behavior is certainly not thermal, indicating the occurrence of non-thermal structural change."

I agree to a certain extent here but the nonthermal effect of concern is peak power dependent multiphoton emission rather than LSPR effects leading to ion displacement. (Also I note that the decay time is similar to thermal dissipation for in vacuum conditions.)

"However, the thermal effect cannot be ignored in light-stimulated phase transition, as latent heat is required to be supplied for phase transition. The plasmonic heating of CuS NCs may therefore harness the phase transition in our systems. We plan to conduct a detailed experiment in the future to investigate and clarify the role of plasmonic heating."

I highly recommend the authors do the peak power dependence by varying the pulse duration under constant fluence (within the limits of Landau damping) and most important look for surface photoemission. I am sure this effect is operating. It is only a question about how big an effect this has on the lattice potential. I can well imagine this being a much bigger effect than any thermal process (even with extremely high surface temperatures in the surface plasmon region...as this too will lead to thermoionic emission).

Answer: According to the comment, we performed the time-resolved electron diffraction measurements by changing the excitation pulse duration (without changing the fluence) (Figure R1). As mentioned above, the intensity changes in shorter and longer pulse durations were identical, suggesting that the effect of the surface photoemission is not strong compared to that of the photoinduced atomic motions. This is logical because the structural changes occur even under CW light, as shown in Figure 2a of the main manuscript.

Action: We added Figure R1 as Figure S10 to the revised version of the Supplementary Information. Additionally, we added a discussion regarding multiphoton excitation on page 11, third paragraph of the revised version of the manuscript.

Please note, I strongly support publication of this work. I hope the authors will do the above controls. I apologize if I was not clear enough in my original report to bring this issue to their full attention. I think a compromise could be made in which the authors estimate the magnitude of the surface fields in the presence of the LSPR and degree of photoemission. It would be instructive for them to look at fs studies of photoemission due to surface plasmons of Cu, some of the early published work on this topic.

Answer: We appreciate your comments, suggestions, and strong support, which enriched our manuscript. We believe that the revised manuscript with additional experiments effectively shows the photoexcitation through the LSPR band inducing the atomic motions in CuS NCs. We are also grateful for your suggestions, which significantly improved the quality of our manuscript.

Reviewer #2 (Remarks to the Author):

The authors have carefully considered my comments and suggestions. I hence suggest acceptance of the manuscript in its present form.

Answer: We appreciate the recommendation of our manuscript to be published on *Nature Communications*. We are also thankful for providing us an opportunity to discuss the mechanism of LSPR stimulating ionic displacement of CuS NCs.

Reviewer #3 (Remarks to the Author):

Through the revision process, the authors clearly answer with sufficient explanations as follows and data sets for suggested questions:

The time scale of the plasmon excitation and relaxation process are explained through schematics and proper reference.

It is properly answered through the SAED pattern that the ionic displacement corresponds to the cooperative phenomenon and that the LSPR excitation of CuS does not change in the long-range period.

Regarding the photoconductivity experiment, it is answered faithfully by presenting data sets where wavelength corresponding to the inside and outside of the two values previously presented. (400 and 800 nm)

Answer: We appreciate your comment on our response. We are also thankful for the important indications and suggestions, which has significantly improved our manuscript.

REVIEWERS' COMMENTS

Reviewer #1 (Remarks to the Author):

The paper is now acceptable for publication.